# ROYAL SOCIETY
# OPEN SCIENCE

Subject Areas:
psychology/cognition

Keywords:
cognitive control, Stroop effect, post-hypnotic suggestion, higher order thoughts, metacognition

Author for correspondence:
B. Palfi
e-mail: palfibence@gmail.com

# Strategies that reduce Stroop interference

B. Palfi[2], B. A. Parris[3], A. F. Collins[1] and Z. Dienes[1,2]

[1]School of Psychology, University of Sussex, Brighton, UK
[2]Sackler Centre for Consciousness Science, University of Sussex, Pevensey Building 1, North South Road, Brighton, East Sussex BN1 9QH, UK
[3]Department of Psychology, University of Bournemouth, Poole, UK

 BP, 0000-0002-6739-8792; BAP, 0000-0003-2402-2100; ZD, 0000-0001-7454-3161

A remarkable example of reducing Stroop interference is provided by the word blindness post-hypnotic suggestion (a suggestion to see words as meaningless during the Stroop task). This suggestion has been repeatedly demonstrated to halve Stroop interference when it is given to highly hypnotizable people. In order to explore how highly hypnotizable individuals manage to reduce Stroop interference when they respond to the word blindness suggestion, we tested four candidate strategies in two experiments outside of the hypnotic context. A strategy of looking away from the target words and a strategy of visual blurring demonstrated compelling evidence for substantially reducing Stroop interference in both experiments. However, the pattern of results produced by these strategies did not match those of the word blindness suggestion. Crucially, neither looking away nor visual blurring managed to speed up incongruent responses, suggesting that neither of these strategies is the likely underlying mechanism of the word blindness suggestion. Although the current results did not unravel the mystery of the word blindness suggestion, they showed that there are multiple voluntary ways through which participants can dramatically reduce Stroop interference.

## 1. Introduction

An essential feature of the human cognitive system is its ability to attend to and use goal-related stimuli while ignoring the distractors of the environment. The Stroop task ([1]; for a review see [2]) provides a window into selective attention and since its publication, it has inspired many theories of attention and cognitive control [3–7]. This task requires participants to name the displayed colour of the presented words while they disregard the meaning of the words. People produce the quickest responses on congruent trials in which the meaning of the presented word is in accordance with its displayed colour (e.g. RED displayed in *red*), followed by the neutral trials in which the meaning of the presented words is unrelated to

colours (e.g. LOT displayed in *red*). The slowest response times (RTs) can be observed on incongruent trials where the displayed colour and the meaning of the words are not in harmony (e.g. RED displayed in *blue*). Performance on the task can be assessed by computing RT differences between these experimental conditions. The Standard Stroop effect is the RT difference between incongruent and congruent trials, and it can be broken down into two components; namely, the Stroop interference effect, which is the RT difference of incongruent and neutral trials, and the Stroop facilitation effect, which is the RT difference of the neutral and congruent trials.

The Stroop effect is remarkably large, and many report experiencing cognitive conflict during an incongruent trial [2]. A long line of research has demonstrated that the presence of the Stroop effect is very robust; it persists despite long-term training (e.g. [8]), and bringing it under control through the application of deliberate strategies is difficult [2]. While methods have been reported that result in reduced Stroop effects [9–12] all involve a manipulation of the stimulus context (e.g. colouring a single letter instead of all letters or decreasing the response–stimulus interval) so as to provide exogenous support to control mechanisms, and are thus not likely the consequence of deliberate, top-down control. Even financial rewards offered to increase motivation to perform well result in either possibly no effect on reaction times other than a general speeding up on all trial types [13,14] or only small (approx. 10 ms) reductions of the Stroop effect [13].

One of the few exceptions to the robustness of the Stroop effect may be provided by the word blindness post-hypnotic suggestion [15,16]. When the word blindness suggestion, a suggestion to see the words during the Stroop task as gibberish or meaningless characters, is given to highly hypnotizable people (henceforth highs), they can substantially reduce the Stroop effect compared with a standard, no suggestion condition. This finding has been replicated by the original authors as well as across independent laboratories (e.g. [16–21]). The magnitude of Stroop interference in the suggestion condition is roughly half the size of the effect in the no suggestion condition (for a meta-analysis, see table 1 of [22]). By contrast, the influence of the word blindness effect on the facilitation component of the Stroop effect (neutral RT minus congruent RT) appears to be more volatile. Importantly, responding to the suggestion speeds up RTs during incongruent trials compared with the no suggestion condition, as compared with the control group of low hypnotizable people. Hence, the effect is an interesting use of cognitive control that is not produced simply by holding back and so slowing down on neutral and congruent trials (thereby equalizing RTs on all trials; [23]).

Thus, the question arises: what exactly happens when highs respond to this post-hypnotic suggestion? Many of the theories of hypnosis concur that responding to a hypnotic suggestion involves top-down cognitive control processes and that the feeling of involuntariness, which is the central feature of the hypnotic phenomena [24,25], is only the result of a deteriorated or relinquished metacognition ([26,–30]; for a review see [31]).[1] Perhaps, the simplest theory of hypnosis (i.e. operates with the fewest assumptions) is the cold control theory, which takes reduced metacognition as the fundamental process of hypnotic responding. Specifically, it asserts that hypnotic responding is implemented by intentional control. Subjects intentionally engage in perceptual or cognitive strategies to create the experiences described in the suggestion while they are able to alter their monitoring over their intentions and make themselves believe that they are not acting deliberately [27,36,37]. The theory draws on the higher order thought (HOT) theories of consciousness [38,39], according to which a mental state becomes conscious by virtue of a higher order state referring to it. For instance, to create the experience of a buzzing mosquito, one can form the following first-order intention: 'imagine a buzzing mosquito'. To be aware that one is engaged in imagination, one would need a second-order state that refers to the first-order state (i.e. 'I intend to imagine a buzzing mosquito'). One can also create the experience of this noise without being aware of the first-order intention (i.e. 'I do not intend to imagine a buzzing mosquito'), and in that case it would feel as if it happened by itself, akin to the experience of hallucination. Importantly, this experience of involuntariness is what hypnotic subjects report about their behaviour when they respond to suggestions. Taken together, according to cold

[1]One exception to this is the response expectancy theory [32,33], which provides a simple explanation of hypnotic responding that does not involve altered metacognitive processes. The theory postulates that expectations, produced by hypnotic suggestions, are enough by themselves to create the experiences and behaviour of hypnotic subjects. The subjects feel these responses are involuntary due to the processes being truly unintentional, as there is no need to involve intentional cognitive control processes. This theory is not mutually exclusive with the theories involving cognitive control and metacognitive processes. However, measured expectations do not fully account for hypnotic responding [34,35]. These findings may be due to measure unreliability but they also give rise to alternative accounts such as the metacognitive theories of hypnotic responding. Therefore, in this paper we focus on the explanation and predictions of the metacognitive theories to understand the underlying mechanism of the word blindness suggestion.

control theory, responding to a suggestion consists of engaging in a strategy to produce the experience described in the suggestion without being aware of using a strategy. From this assumption, it follows that the sole difference between a hypnotic and a non-hypnotic response is the form of the accompanying second-order state. Therefore, if one is capable of reducing the Stroop interference effect by responding to the word blindness suggestion, one should be able to do it by voluntary, non-hypnotic means as well, using the very same strategy that they used when they responded to the suggestion. Identifying such a strategy is central to cold control theory and to simple, metacognitive explanations of hypnosis, as the lack of a clear explanation involving intentional actions invites more complex theories to address the word blindness suggestion.

We review four unique strategies here that have the potential to be regarded as an underlying mechanism of the word blindness suggestion. Some of these strategies were reported to be spontaneously used by highs (and lows) when they undertook the Stroop task outside of the hypnotic context [40], and some are simple strategies that have the face validity to be able to reduce Stroop interference. The most straightforward candidate is the looking-away strategy. Subjects may divert their attention from the word so that they can easily process the colour but not the meaning of the word, which can result in a reduced interference. Indeed, it has been demonstrated that lows can reduce the Stroop interference by diverting their attention from the words [19]. However, Raz et al. [15,19] argued that it is unlikely that highs engage in this strategy when they respond to the suggestion. For example, subjects first reported that they observed words in all instances and that they claimed that they did not engage in any spatial attentionally related strategies. Second, the experimental sessions were videotaped, and independent judges were unable to distinguish between highs and lows based on their eye-movement patterns. Nonetheless, these arguments are far from bulletproof. As stated earlier, it is the essence of hypnosis that when subjects respond hypnotically, they can engage in strategies without being aware of doing so [27,30]; hence, asking them whether they used any strategies may not be a sensitive way to explore the underlying mechanism of the suggestion. Moreover, human judges may not be able to observe eye-movement patterns, and thus an objective criterion based on, for instance, the fixation time outside of the area of interest defined around the words could provide a more severe test of the strategy.

A more subtle form of the looking-away strategy is when subjects focus their attention toward a single letter, or a portion of a letter so that they can more easily name the font colour. There is ample evidence that colouring only the last or the first letter of a Stroop word compared with the middle letter decreases the size of the Stroop interference effect ([10,12,41]; for a review see [42]). Moreover, highs can respond more quickly during incongruent trials when this strategy is provided in a hypnotic context ([40]; cf. [43]). Nonetheless, the Sheehan et al. [40] study lacked a non-hypnotic strategy condition; hence, it is unclear whether the inclusion of hypnosis in the strategy condition increased the motivation and expectations of highs compared with the non-hypnotic baseline condition. Thus, the lack of appropriate control could create a 'hold-back' effect [30,44] in the non-hypnotic baseline condition as a way of satisfying demand characteristics.

Another visually related strategy is blurring. Subjects may adjust visual accommodation (e.g. by relaxing of the muscles around their eyes) so that the image of the word does not fall directly on the retina. Blurring may prioritize the colour of the word over the meaning. Raz et al. [19] provided a test of this strategy by administering a pharmacological agent to highs to disrupt visual accommodation (in other words, induce the state of cycloplegia). The subjects were exposed to two drops of 1% cyclopentolate hydrochloride and their vision was corrected by lenses so that they saw the words clearly during the Stroop task. Remarkably, highs still decreased the Stroop interference effect when they responded to the suggestion compared with the no suggestion condition. One might, therefore, conclude that highs achieved the reduction by means other than visual blurring. However, this conclusion is conditional on the participants being in a state of complete cycloplegia. There was no outcome neutral test examining whether the participants had completely lost their ability to accommodate vision. The authors point out that residual accommodation can still occur, especially for younger participants, when this particular agent is used.

Finally, there is evidence that subjects spontaneously resort to a strategy that involves the rehearsal of the task instructions, such as the word 'colour' [40]. Goal maintenance has been shown to play a critical role in task performance in the Stroop task, therefore, a strategy that sustains an active goal representation might help participants mitigate Stroop interference [9,45,46].

The purpose of this project is to explore the underlying mechanism or mechanisms of the word blindness suggestion by testing whether any of these four strategies (looking-away, visual blurring, single-letter focus and goal-maintenance) could be one that highs use when they respond to the

suggestion.[2] The test of these strategies is especially relevant to the cold control theory of hypnosis, as it expects that suggestions are implemented by intentional actions, but it also has the potential to further our understanding of the Stroop task and cognitive control. To test the efficiency of the strategies, we designed a fully within-subjects experiment in which participants undertook the Stroop task in five separate blocks: in four blocks they were explicitly asked to use one of the mentioned strategies and in one block they were told to not use any of these strategies (baseline/control condition). According to the cold control theory, if a strategy can be applied hypnotically to reduce the Stroop effect, it should be equally available and applicable non-hypnotically. Hence, the experiment was administered outside of the hypnotic context; in fact, no reference was made to hypnosis or to the word blindness suggestion. The key tests were whether each strategy could reduce Stroop interference, and whether the reduction happens via speeding up RTs of incongruent trials. We did not define a key test involving Stroop facilitation as the effect of the word blindness suggestion on this component is unclear [22]. In order to allow for the comparison of our results and the results of earlier studies demonstrating the word blindness effect, the stimuli and design (e.g. manual version of the Stroop task) of our experiments were largely the same as those of the original study of Raz *et al.* [15].

As a secondary analysis, we tested whether the efficiency of a specific strategy is related to hypnotizability beyond the effect of expectations and motivations conditional on a hypnotic context. Cold control theory postulates that individual differences in hypnotizability are grounded in differential metacognitive skills (which may or may not be limited to the domain of intentions) and not in differential cognitive control. Empirical evidence is in harmony with this assumption [48–50]. Consequently, lows and mediums should be able to use a specific strategy just as efficiently as highs, when they are sufficiently motivated. This assumption also has practical relevance to the current study, as it implies that to test the strategies recruitment does not need to be limited to highs only. Nonetheless, if the results reveal a positive relationship between hypnotizability and strategy efficiency outside of the hypnotic context, the purely metacognitive account of hypnosis would need to be revised, and the plausible strategies would need to be tested only on highs. To exclude the effect of expectations and motivations regarding hypnosis, we recruited participants from a subject pool where the majority of the people had already been screened for hypnotizability, so that we would not need to disclose the hypothesis to the participants. Consent to link results to hypnotizability scores was acquired after the experiment; therefore, it is unlikely that they could associate the current experiment in any way with hypnosis or hypnotizability.

# 2. Experiment 1

## 2.1. Methods

### 2.1.1. Participants

We recruited 78 participants from which 57 (mean age = 19.61, s.d. = 1.47, females = 51) had been screened for hypnotizability with the Sussex-Waterloo Scale of Hypnotizability (SWASH; [51]). As we specified in the pre-registration, we excluded the data of those who did not have a SWASH score from all of the analyses. The experiment was advertised for first- and second-year psychology students of the University of Sussex who finished a module earlier in which they had the opportunity to participate in a hypnosis screening session. High and low hypnotizability were defined as scoring in the top and bottom 15% of the SWASH, respectively. We calculated the cut-off *a priori* based on the composite (objective and subjective) SWASH scores of all the first- and second-year students in our database. The cut-off for highs was 5.35 whereas the cut-off for lows was 2.00 (on a scale of 0 to 10). From the 57 participants, 10 were high, 39 medium and 8 low hypnotizables. The participants were proficient readers of English and they attended the experiment in exchange for course credits. All participants gave their informed consent before the experiment as well as after the experiment when we revealed that we wished to correlate their performance with their hypnotizability scores. The Ethical Committee of the University of Sussex approved the study (ER/BP210/5).

---

[2]Note that there may be a fifth strategy. It may be that highly hypnotizable people use the instructions of the word blindness suggestion as a strategy and by creating the experience of meaninglessness they can reduce Stroop interference. We tested this possibility in another paper of ours [47] and we discuss its results and implications in the General discussion.

## 2.1.2. Stimuli and apparatus

The current stimuli closely followed those used by Raz *et al.* [15] for the purpose of comparability. The stimulus set included four types of colour words (RED, BLUE, GREEN and YELLOW) and four types of neutral words (LOT, SHIP, KNIFE and FLOWER). The congruent trials consisted of colour words presented in colours matching the meaning of the words (e.g. RED in the colour red). The incongruent trials were colour words displayed in colours mismatching the meaning of the word (e.g. RED in the colour blue) covering all possible pairings of presented colours and meanings. The colour and the neutral words were frequency and length matched. All words were written in upper-case font and presented against a white background. The words were presented in the following hex colour codes: #ff0000 (red), #0000ff (blue), #008000 (green) and #ffef36 (yellow). The vertical visual angle of the stimuli was 0.5°, while the horizontal visual angle of the stimuli lie between 1.3° and 1.9° depending on the length of the word. The distance between the participants' eyes and the computer screen was approximately 65 cm. The response keys used in the experiment were 'V', 'B', 'N' and 'M' for the colours red, blue, green and yellow, respectively. The keyboard buttons were not colour labelled (note that Raz *et al.* [15] used colour labels; however, we did not provide these visual aids to control for a potential colour-matching strategy). The experiment was produced in and run by the software OpenSesame [52] on a computer with a screen resolution of 1366 × 768 (15.6-inch screen).

## 2.1.3. Design and procedure

The study had a 3 × 5 × 3 mixed design with the independent variables of the congruency type of the trial (congruent versus neutral versus incongruent), the strategy used in the conditions (no strategy, looking away, blurring, single-letter focus, goal-maintenance) and hypnotizability (low, medium or high).[3] The proportion of congruent, neutral and incongruent trials was equal (33%) in each. The order of conditions as well as the order of the Stroop trials (144 per condition) were randomized across participants.

The experiment took place in a dimly lit room with the experimenter present and only one participant at a time. The participants were told that they will undertake the Stroop task several times and, in some cases, they will be provided with explicit instructions to use a specific strategy to help them with the task. After providing their informed consent to the study, the participants engaged in a practice Stroop task (36 trials). The participants were instructed to place their left middle finger on 'V', left index finger on 'B', right index finger on 'N' and right middle finger on 'M' while undertaking the Stroop task. They were asked to respond to the colour of the word on the screen as quickly and as accurately as they can. The participants were instructed to focus on the fixation cross and retain their focus on the centre of the screen during the Stroop task. After 1500 ms, the fixation cross was replaced by one of the Stroop words and remained on the screen until a response was given or for 2000 ms. Finally, a feedback (CORRECT or INCORRECT) flashed in black on the screen and then a new trial started with the fixation cross. The response to stimulus interval was 2000 ms. This sequence remained constant across all conditions.

Next, the participants undertook the five experimental conditions. The order of the conditions was randomly generated for each participant. In the no strategy condition, the participants were asked to not use any of the mentioned strategies, and to respond as fast and as accurately as they could. All strategy conditions started with a screen explaining the strategy they are asked to use on each trial. For the visual strategies, an example word was presented so that the participants could practice the strategy (see the appendix for exact instructions). Before the start of the condition, the experimenter asked the participants whether they had understood how to use the strategy and provided clarification on request. After each strategy condition, the participants were asked to report the percentage of the trials on which they managed to use the strategy. (What do you think, on what percentage of the trials did you use the strategy? Please answer with a number between 0 and 100.) After finishing the last condition, the participants were thanked and debriefed.

## 2.2. Data analysis

### 2.2.1. Statistical analyses

We conducted all of our analyses with the statistical software R 3.3.1 [53]. We calculated difference scores for the RTs so that we were able to directly test all of our hypotheses with Bayesian paired *t*-tests

---

[3]Note that hypnotizability was measured as a continuous variable, and we created groups using cut-offs described in the Participants subsection of the Methods section.

(comparing two conditions or testing whether a regression slope is different from zero) or Bayesian independent *t*-tests. Note that we did not run any omnibus tests (e.g. F test including all five conditions at a time) as it would not be informative in respect of hypotheses of the current project. We reported *p*-values for each statistical test, but we used the Bayes factor (B) to draw conclusions.

### 2.2.2. Bayes factor

We applied the R script of Dienes & McLatchie [54] to calculate the Bayes factors. This calculator has a *t*-distribution as a likelihood function for the data as well as for the model of H1. We set the degrees of freedom of the model of H1 to 10 000 in each analysis to have a likelihood function for the theory following a normal distribution. To calculate the B, one also needs to specify the prediction of the two models (H1 and H0) under comparison. Every tested hypothesis had directional prediction; hence, we applied a half normal distribution with a mode of zero to model the predictions of H1. We specified the distribution as a half-normal since it is in line with the assumption that smaller effects are more probable than larger effects [55]. We report Bs as $B_{H(0,X)}$, in which H indicates that the model is half-normal, the first parameter (0) indicates the mode of the distribution and the second parameter (X) represents the s.d. of the distribution. We used various strategies to define the s.d.s of the different H1s.

Concerning the outcome neutral tests of the Stroop interference and the Stroop effects, we informed the s.d. of the models predicting these effects based on the results of the baseline condition of a recent study of ours that used identical Stroop materials [47]. That is, the s.d. of the models of the Stroop interference and Stroop effects were 60 and 105 ms, respectively. For the critical analysis, testing the efficiency of the strategies, we used 30 ms, which is half of the baseline Stroop interference. This value is based on the finding that the word blindness suggestion usually halves the baseline Stroop interference and we expect that a successful strategy should produce about the same effect size [22]. Incidentally, this value is exactly the same as we would obtain by using the room-to-move heuristic to define the maximum possible effect size, provided that the baseline Stroop interference is 60 ms [56]. The s.d. of the model predicting a positive relationship between hypnotizability and reduction in Stroop interference by strategy application was 5 ms, and it was based on the findings of Parris & Dienes [18], who demonstrated a positive link between hypnotizability and the imaginative word blindness effect. In other words, H1 predicts that one unit increase on the SWASH aids the ability to reduce the Stroop interference using one of the strategies with about 5 ms.

In order to draw conclusions about the compared models, we used the convention of B > 3 to distinguish between insensitive and good enough evidence for the alternative hypotheses [57]. By symmetry, we used the cut-off of B < 1/3 to identify good enough evidence for the null hypothesis. To evaluate the robustness of our Bayesian conclusions to the s.d.s of the H1 models, we report a robustness region for each B, providing the range of s.d.s of the half-normal models that qualitatively support the same conclusion (using the threshold of 3 for moderate evidence for H1 and ⅓ for moderate evidence for H0) as the chosen s.d. [56,58. The robustness regions are reported as: RRconclusion x1 x2] where x1 is the smallest and x2 is the largest s.d. that gives the same conclusion: B < 1/3, 1/3 < B < 3, B > 3.

## 2.3. Pre-registration

The design and analysis plan of this experiment was pre-registered at https://osf.io/4z3xu. We closely followed the steps of the pre-registration when running the experiment and the analysis. Nonetheless, we added an analysis to the set of the crucial tests (Crucial test 1): the test of the efficiency of the strategies with all participants who had SWASH scores. This analysis is critical to demonstrate whether or not there is a main effect of successful strategy application irrespective of the participants' hypnotizability.

## 2.4. Results

### 2.4.1. Data processing

We excluded the trials with errors from the analyses (8.2% in total, of which 1.3% were from the no strategy, 2.1% from the looking away, 1.6% from the blurring, 1.7% from the single letter focus and 1.5% from the goal-maintenance conditions).[4] Following the outlier exclusion criterion of Raz *et al*.

---

[4]See electronic supplementary material for the analyses of the error rates (Exploration S5).

**Table 1.** Summary table about the means of the RTs (ms) in the five strategy conditions. Note: the standard deviations (s.d.) of the means are shown within the brackets.

| strategy condition | congruency type | | |
| --- | --- | --- | --- |
| | incongruent | neutral | congruent |
| no strategy | 808 (127) | 730 (101) | 682 (94) |
| looking-away | 815 (94) | 802 (94) | 771 (97) |
| blurring | 821 (121) | 776 (119) | 739 (114) |
| single-letter focus | 880 (157) | 812 (133) | 766 (130) |
| goal-maintenance | 804 (142) | 726 (107) | 689 (90) |

[15], we omitted trials with RTs that were three standard deviations either above or below the mean. The proportions of outliers were low and comparable across conditions (we excluded 1.2% of the correct trials, of which 0.2% were from the no strategy, 0.3% from the looking away, 0.3% from the blurring, 0.2% from the single letter focus and 0.2% from the goal-maintenance conditions).

### 2.4.2. Outcome neutral checks 1 (non-preregistered): on what percentage of the trials did the participants use the strategies?

The conditions in descending order based on the means of the reported percentages of strategy usage: goal-maintenance ($M = 86\%$, 95% CI [82%, 90%]); looking away ($M = 83\%$, 95% CI [80%, 87%]); blurring ($M = 73\%$, 95% CI [68%, 78%]); and single-letter focus conditions ($M = 66\%$, 95% CI [61%, 71%]).

### 2.4.3. Outcome neutral tests 2: is there a Stroop interference effect in the no strategy condition?

As anticipated, the RTs in the no strategy condition were the fastest in the congruent trials followed by the neutral trials and then the incongruent trials (table 1 for condition means and s.d.s). The comparison of the incongruent and neutral trials yielded evidence for the Stroop interference effect ($t_{56} = 7.74$, $p < 0.001$, $M_{\text{diff}} = 78$ ms, $d_z = 1.03$, $B_{H(0,60)} = 1.49 \times 10^8$, $RR_{B > 3}[3, 2.76 \times 10^4]$). The contrast of the incongruent and congruent trials revealed evidence in support of the Stroop effect ($t_{56} = 11.73$, $p < 0.001$, $M_{\text{diff}} = 126$ ms, $d_z = 1.55$, $B_{H(0,105)} = 2.23 \times 10^{14}$, $RR_{B > 3}[4, 4.62 \times 10^4]$).

### 2.4.4. Crucial test 1 (non-preregistered): are the strategies effective in reducing the Stroop interference effect?

Using the data of all the participants we tested whether any of the four strategies decreased Stroop interference (incongruent RTs—neutral RTs). Comparing the no strategy and strategy conditions revealed evidence for the effectiveness of the looking-away ($t_{56} = 4.99$, $p < 0.001$, $M_{\text{diff}} = 65$ ms, $d_z = 0.66$, $B_{H(0,30)} = 3.93 \times 10^3$, $RR_{B > 3}[5, 2.05 \times 10^4]$) and the blurring ($t_{56} = 2.85$, $p = 0.006$, $M_{\text{diff}} = 33$ ms, $d_z = 0.38$, $B_{H(0,30)} = 20.05$, $RR_{B > 3}[6, 365]$) strategies. There was anecdotal evidence for no difference between no strategy and the single-letter focus ($t_{56} = 0.73$, $p = 0.469$, $M_{\text{diff}} = 9$ ms, $d_z = 0.10$, $B_{H(0,30)} = 0.73$, $RR_{1/3 < B < 3}[0, 74]$), and between the no strategy and goal-maintenance strategies ($t_{56} = 0.01$, $p = 0.993$, $M_{\text{diff}} = 0$ ms, $d_z = 0.00$, $B_{H(0,30)} = 0.38$, $RR_{1/3 < B < 3}[0, 34]$). The Bayes factor of the latter two tests did not reach the level of good enough evidence. See figure 1 for the distribution of the Stroop interference scores and table 1 for congruency condition means and s.d.s broken down by the strategy conditions.

### 2.4.5. Crucial test 2 (non-preregistered): do the strategies decrease the RTs of the incongruent trials?

Interestingly, the mean RTs of incongruent trials in the looking-away and blurring conditions were numerically higher than that of the no strategy condition. We found evidence that neither the looking-away ($t_{56} = -0.46$, $p = 0.647$, $M_{\text{diff}} = -7$ ms, $d_z = -0.06$, $B_{H(0,30)} = 0.34$, $RR_{1/3 < B < 3}[0, 30]$) nor the blurring strategies ($t_{56} = -0.86$, $p = 0.392$, $M_{\text{diff}} = -13$ ms, $d_z = -0.11$, $B_{H(0,30)} = 0.27$, $RR_{B < 1/3}[23, \infty]$) reduced the RTs of incongruent trials. Bayesian evidence regarding the slow-down of incongruent RTs remained insensitive for both the looking-away ($B_{H(0,30)} = 0.65$, $RR_{1/3 < B < 3}[0, 66]$) and the blurring strategies ($B_{H(0,30)} = 0.93$, $RR_{1/3 < B < 3}[0, 101]$).

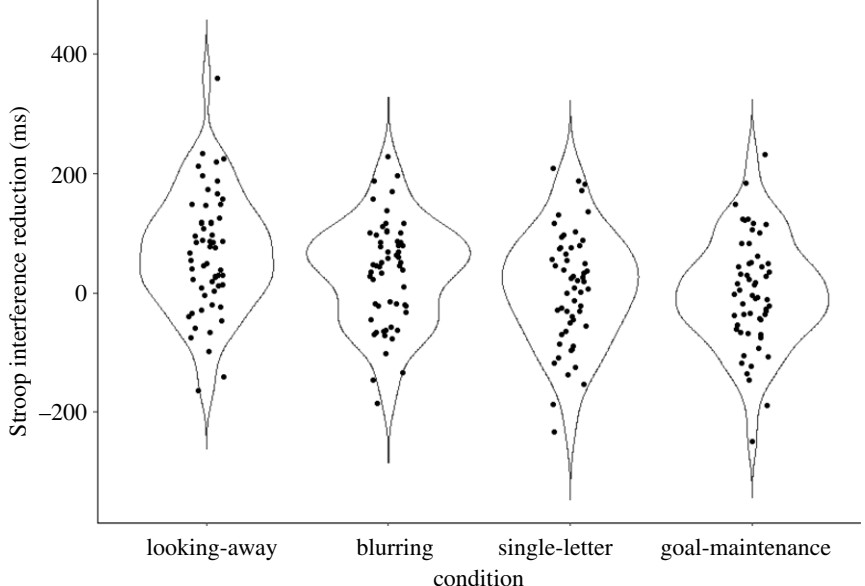

**Figure 1.** Violin plot depicting the distribution of Stroop interference score differences (ms) between the no strategy and the four strategy conditions. Each black dot represents the reduction of the Stroop interference score (incongruent RT—neutral RT) by a specific strategy of a single participant.

### 2.4.6. Crucial test 3: is there a relationship between hypnotizability and the extent to which people can reduce the Stroop interference by the tested strategies?

To this aim, we regressed the SWASH scores on the extent of the reduction in Stroop interference by the strategies and tested the regression slopes against zero. Even though the raw regression slopes are comparable to zero, we did not gain good enough evidence for the null in any case. The raw regression slopes in descending order: blurring ($t_{55} = 0.25$, $p = 0.801$, $b = 1.74$ ms/SWASH unit, $\beta = 0.03$, $B_{H(0,5)} = 0.91$, $RR_{1/3 < B < 3}[0, 24]$), single-letter focus ($t_{55} = 0.11$, $p = 0.920$, $b = 0.79$ ms/SWASH unit, $\beta = 0.01$, $B_{H(0,5)} = 0.92$, $RR_{1/3 < B < 3}[0, 23]$), looking-away ($t_{55} = 0.06$, $p = 0.950$, $b = 0.49$ ms/SWASH unit, $\beta = 0.01$, $B_{H(0,5)} = 0.86$, $RR_{1/3 < B < 3}[0, 23]$) and goal-maintenance strategy ($t_{55} = -0.11$, $p = 0.911$, $b = -0.81$ ms/SWASH unit, $\beta = -0.2$, $B_{H(0,5)} = 0.78$, $RR_{1/3 < B < 3}[0, 18]$). Figure 2 depicts the scatterplots, regression slopes and their 95% confidence intervals for each strategy separately. The electronic supplementary material reports an alternative analysis of this question in which we directly compared the group of highs and lows in the extent to which they reduced the Stroop interference effect. Importantly, the results are in accordance across the analyses.

### 2.4.7. Supporting test of interest 1 (non-preregistered): do the strategies influence RTs in general?

Table 1 suggests that the strategies may trigger a general slow-down effect on RT. To test this, we ran four Bayesian $t$-tests comparing the average of the incongruent and neutral RTs of the no strategy and every strategy conditions. Note that these analyses are equivalent to four tests of the main effect of each strategy on the RTs of incongruent and neutral trials. We found good enough evidence for a general slow-down effect for the looking-away ($t_{56} = 3.20$, $p = 0.002$, $M_{diff} = 40$ ms, $d_z = 0.40$, $B_{H(0,30)} = 43.33$, $RR_{B>3}[6, 992]$), the blurring ($t_{56} = 2.42$, $p = 0.019$, $M_{diff} = 29$ ms, $d_z = 0.26$, $B_{H(0,30)} = 8.34$, $RR_{B>3}[8, 132]$) and the single-letter focus strategies ($t_{56} = 2.42$, $p = 0.019$, $M_{diff} = 29$ ms, $d_z = 0.26$, $B_{H(0,30)} = 8.34$, $RR_{B>3}[8, 132]$). By contrast, goal-maintenance did not increase the RTs to incongruent and neutral trials ($t_{56} = 0.39$, $p = 0.700$, $M_{diff} = -4$ ms, $d_z = -0.04$, $B_{H(0,30)} = 0.26$, $RR_{B>3}[23, \infty]$).

## 2.5. Discussion

In this experiment, we tested four strategies that putatively reduce the Stroop interference effect to examine whether any of these strategies can be the underlying mechanism of the word blindness suggestion. The crucial test of the strategies provided insufficient evidence either way for whether the single-letter focus or the goal-maintenance strategies could mitigate the extent of the interference. On the other hand, the looking-away and the visual blurring strategies passed the crucial tests as they

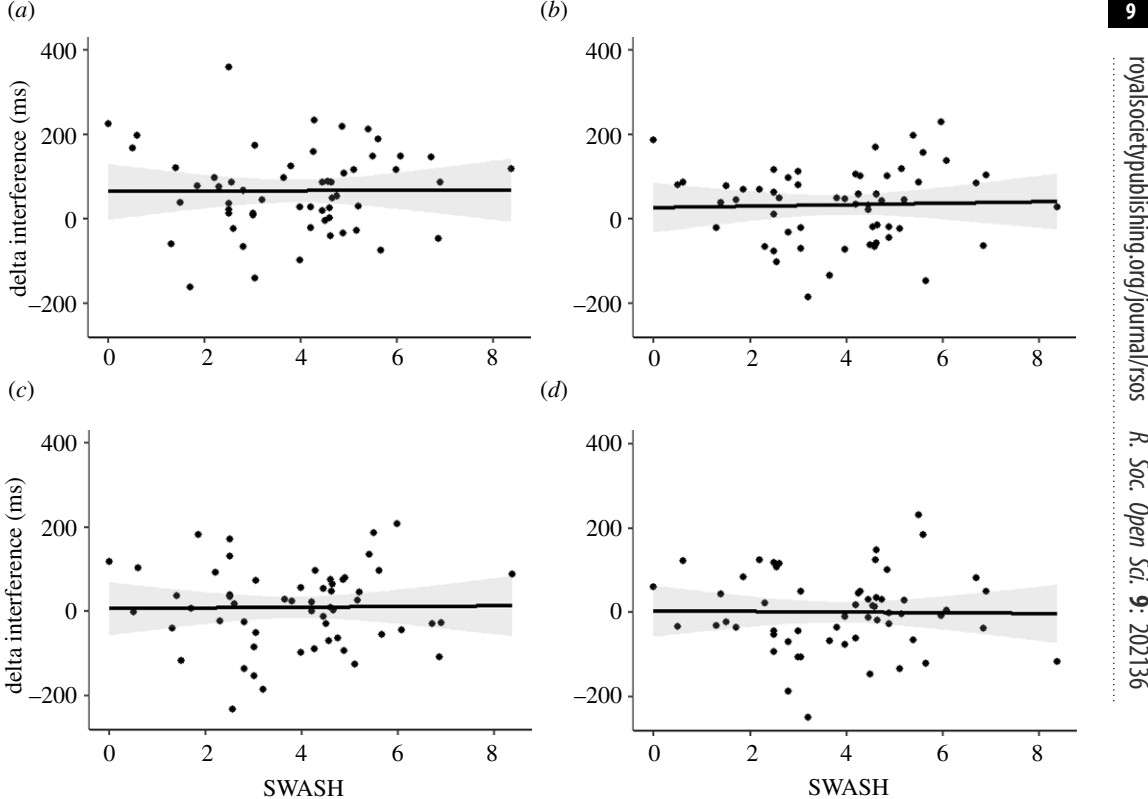

**Figure 2.** Scatterplots showing the relationship between hypnotizability (measured by the SWASH) and the reduction in the Stroop interference induced by the four strategies. The four panels indicate the looking-away (*a*), blurring (*b*), single-letter focus (*c*) and goal-maintenance (*d*) strategies.

drastically decreased the extent of interference for all levels of hypnotizability. Moreover, the blurring strategy approximately halved the extent of Stroop interference (reduction of 33 ms from the baseline of 78 ms), which is precisely what the word blindness suggestion achieves in general [22]. However, as mentioned earlier, the word blindness effect has another distinctive feature: it realizes the reduction of the interference effect by reducing the RTs of incongruent trials ([23]; see electronic supplementary material, table S1 for a meta-analysis of studies demonstrating the word blindness effect and the reduction of incongruent RTs in the suggestion compared with the no suggestion conditions). Surprisingly, our results do not match this pattern; there is evidence that neither of the strategies managed to decrease the RTs of the incongruent trials (for the looking-away strategy as the corresponding B = 0.34 was just above the conventional rough guideline of i < 1/3). If this finding is robust, it challenges the idea that these strategies are the underlying mechanisms of the suggestion. Therefore, in the next experiment, we pre-registered the reduction in incongruent RTs as a test of the strategies. An alternative analysis of this question, in which we compared the strategy conditions with the word-blindness suggestion condition of a different experiment that had the same Stroop materials as the current experiment (see the electronic supplementary material) yielded the same result, namely, that the strategies and the word blindness suggestion produced different patterns.

Another key characteristic of the word blindness suggestion is that it seems to reduce interference by attenuating response competition and not by de-automatizing reading *per se* [17,59]. By introducing colour-associated words (e.g. sky), Augustinova & Ferrand [17] distinguished the effect of the suggestion on the semantic and the response conflict components. Crucially, the de-automatization of reading account predicts the reduction of semantic conflict, whereas the response competition account expects that semantic processing remains unaffected by the suggestion. In two experiments, it was demonstrated that the word blindness suggestion modulated only the *response conflict component* deeming it unlikely that the suggestion operates via the dampening of semantic processing.[5] Parris,

---

[5]Note that the lack of a significant reduction in semantic conflict by the word blindness suggestion can also indicate data insensitivity. Therefore, we calculated Bayes factors comparing evidence for the de-automatization account and against the response competition model. We found a $B_{H(0,10)}$ of 0.42 in Experiment 1 and a $B_{H(0,10)}$ of 0.39 in Experiment 2. Next, we meta-analytically combined

Dienes & Hodgson [22] argued on the basis of response time distributional analysis that the word blindness suggestion took its effect on the portion of the response time distribution associated with response conflict and not semantic conflict (for other behavioural evidence supporting the response competition account, see Palfi, Parris, Seth, & Dienes [60]; cf. the neural correlates of the word blindness suggestion found by [20]. Hence, the strategy that underlies the suggestion should not take its effect by dampening the visual input of the meaning of the words; rather it should aid the subjects to handle response conflict between the competing response options. It is not clear, however, whether looking away or visual blurring would be in accordance with this notion. Therefore, in the next experiment, we introduce a new condition to dissociate the semantic and response conflict components of the Stroop interference effect, and we specify a new crucial test. Namely, a strategy to be deemed a plausible underlying mechanism of the suggestion should only reduce response conflict and should not influence semantic conflict.

# 3. Experiment 2

In this experiment, we aim to test whether the beneficial effects of the looking-away and visual blurring strategies on the mitigation of Stroop interference can be replicated. As argued earlier, the cold control theory assumes that hypnotizability is only related to metacognitive abilities and so strategies used during hypnosis should be applicable to anyone (irrespective of their hypnotizability) inside or outside of hypnosis. As the first experiment did not provide sensitive evidence against this assumption, we retained it and tested the strategies by recruiting participants from the whole range of hypnotizability.

We defined two conditions that the strategies ought to meet to be considered as appropriate underlying mechanisms of the word blindness suggestion: (i) they need to reduce incongruent RTs, and (ii) as suggested by previous findings (e.g. [17]) they should alleviate response conflict rather than semantic conflict. In order to test the latter assumption, we added non-response set incongruent trials to all of the experimental conditions. These trials consist of colour words that are not part of the response set (e.g. brown) displayed in one of the colours of the response set. Therefore, responding to these types of trials should not involve response competition, and the non-response set interference (RT difference between non-response set incongruent and neutral trials) can be taken as an index of conflict that occurs during semantic processing [61,62].[6] Henceforth, we refer to the non-response set interference effect simply as semantic conflict or semantic interference effect.

## 3.1. Methods

### 3.1.1. Participants

We recruited 35 participants; however, one of the participants claimed that they did not follow the instructions closely and used visual blurring in the no strategy condition. We excluded the data of that participant, and all analyses were run on the data of 34 participants (mean age = 21.82, s.d. = 4.38, females = 27). The participants received either course credits or payment (£5) in exchange for attending the study.

### 3.1.2. Stimuli and apparatus

The materials of the registered experiment closely followed those in the first experiment. We added four colour words to the stimulus set (BROWN, PINK, GREY, ORANGE) and created two independent stimulus sets defined by the colours in which the words are presented (A and B). In set A, all words were presented in one of the original colours (red, blue, green or yellow) and so the non-response set

---

evidence from these two experiments and calculated a meta Bayes factor: $B_{H(0,10)} = 0.29$. This implies that we have Bayesian evidence in support of the model predicting no effect on semantic conflict (i.e. the response competition model is supported).

[6]To distinguish between the semantic and response conflict components of the Stroop interference effect, one can also use colour-associated words (e.g. sky) that tend to produce longer RTs than neutral words but shorter RTs than response set incongruent trials [61]. For instance, Augustinova and Ferrand [17] applied colour-associated words in their experiments to assess the magnitude of semantic conflict and to present evidence that the word blindness suggestion influences solely the response conflict component of the interference effect. Nonetheless, their experiments employed vocal responses, and when it comes to manual responses, the colour-associated interference effect is volatile [62,63].

incongruent trials comprised the new colour words presented in the original colours. In set B, all words were presented in one of the new colours (brown, pink, grey or orange) and so the non-response set incongruent trials consisted of the original colour words presented in the new colours. The hex colour codes of the new colures were #a52a2a (brown), #ffaaff (pink), #808080 (grey) and #ffa500 (orange). We ran the experiment in OpenSesame [52] and the resolution of the computer screen was $1920 \times 1080$ (18-inch screen).

### 3.1.3. Design and procedure

There were three major changes in this experiment: we did not include the single-letter and goal-maintenance strategy conditions; there were more trials in each strategy condition as we included non-response set trials as well (we had 48 trials from each trial type, and so 192 trials in total in each strategy condition); we did not take into account the hypnotizability of the participants. The experiment had a $4 \times 3 \times 2$ mixed design with congruency type (congruent, neutral, incongruent non-response set, incongruent response set) and strategy condition (no strategy, looking away, visual blurring), and non-response set groups (response set being equivalent [A] versus not equivalent to the first experiment [B]) as independent variables. The participants were assigned to response set groups A or B based on the parity of their subject number. Group membership determined whether the colours of A or B would have corresponding response buttons. For instance, if someone was assigned to group B, then the colours brown, pink, grey and orange had the corresponding response buttons of 'V','B','N' and 'M', respectively. In this case, none of the words were displayed in red, blue, green or yellow. Apart from this, the procedure of the experiment was identical to that of the first experiment.

## 3.2. Data analysis

The steps of the data analysis are in line with those of the first experiment, including the exclusion criterion regarding RT data and how we drew conclusions based on the results of the Bayes factors. We informed the parameters of the model predicting the presence of the semantic interference effect based on the findings of Augustinova & Ferrand [17], who found in two experiments that the size of the semantic interference (using colour-associated words) was about 20 ms. We expect that an intervention impacting semantic processing should approximately halve this effect. For the test of the regressions slopes investigating the relationship of general response speed and the extent of the Stroop effect, the model parameters of H1 were stemmed from the finding that the slope was 0.13 ms in the no strategy condition in the first experiment. We used this value as the s.d. of H1 for the tests of the slopes against zero as well as for their comparisons.

## 3.3. Pre-registration

The design and analysis plan of the experiment were pre-registered, and they can be accessed at https://osf.io/gbsaf. We closely followed the steps of the design and of the analysis plan.

## 3.4. Results

### 3.4.1. Data processing

First, we omitted trials with errors from the analyses (10.4% in total, of which 2.3% were from the no strategy, 4.4% from the looking away, 3.7% from the blurring conditions). Next, we eliminated trials with RTs that were three standard deviations either above or below the mean. Similarly to the first experiment, the proportions of outliers remained low and comparable across conditions (we excluded 1.2% of all correct trials, of which 0.5% were from the no strategy, 0.4% from the looking-away, 0.3% from the blurring conditions).

### 3.4.2. Outcome neutral checks 1 (non-preregistered): on what percentage of the trials did the participants use the strategies?

The participants reported that, on average, they used on 80% (95% CI [75%, 85%]) of the trials the looking-away strategy, and on 73% (95% CI [66%, 81%]) of the trials the blurring strategy.

**Table 2.** Summary table about the means of the RTs (ms) in the three strategy conditions. Note: the standard deviations (s.d.s) of the means are shown within the brackets.

| | congruency type | | | |
|---|---|---|---|---|
| strategy condition | incongruent | incongruent non-response set | neutral | congruent |
| no strategy | 791 (131) | 746 (112) | 712 (97) | 661 (81) |
| looking-away | 838 (126) | 822 (126) | 830 (127) | 790 (118) |
| blurring | 822 (130) | 812 (130) | 786 (128) | 737 (119) |

### 3.4.3. Outcome neutral tests 2: is there a difference between the two response set groups regarding the magnitude of the Stroop interference and the semantic Stroop effect (in the no strategy condition)?

Before collapsing the data across response set groups, we compared the two groups in terms of the extent of the Stroop interference and semantic Stroop effects. For instance, the colours used in set A were more saturated and luminous than those used in set B, which may made it easier for the participants to differentiate between the response options in the former case. This in turn may have produced a smaller interference or semantic Stroop effect in set A than in set B. The size of the Stroop interference effect was comparable in the two response set groups ($M_A = 78$ ms, $M_B = 79$ ms) and there is weak evidence in favour of the model predicting no difference ($t_{30.66} = -0.05$, $p = 0.958$, $M_{\text{diff}} = 1$ ms, $d_z = 0.02$, $B_{N(0,60)} = 0.38$, $RR_{1/3 < B < 3}[0, 69]$); however, the strength of evidence did not reach the conventional cut-off of good enough evidence. The size of the semantic Stroop effect was numerically larger in the group with the response set of the first experiment ($M_A = 49$ ms, $M_B = 15$ ms); however, the analysis yielded data insensitivity ($t_{29.46} = 1.27$, $p = 0.212$, $M_{\text{diff}} = 35$ ms, $d_z = 0.44$, $B_{N(0,20)} = 1.08$, $RR_{1/3 < B < 3}[0, 179]$). Consequently, we decided to conduct all of the subsequent analyses on the collapsed data.

### 3.4.4. Outcome neutral tests 3: is there a Stroop interference and a semantic Stroop effect in the no strategy condition?

As in the first experiment, the RTs in the no strategy condition were the fastest in the congruent trials followed by the neutral trials. The RTs of the non-response set incongruent trials were slower than those of the neutral trials, and the longest RTs were observed in the incongruent trials (see table 2 for condition means and s.d.s). The analyses revealed strong evidence for Stroop interference ($t_{33} = 6.56$, $p < 0.001$, $M_{\text{diff}} = 79$ ms, $d_z = 1.12$, $B_{H(0,60)} = 2.48 \times 10^5$, $RR_{B > 3}[5, 2.6 \times 10^4]$) as well as for the Stroop effect ($t_{33} = 10.16$, $p < 0.001$, $M_{\text{diff}} = 130$ ms, $d_z = 1.74$, $B_{H(0,105)} = 3.36 \times 10^9$, $RR_{B > 3}[6, 4.57 \times 10^4]$). Moreover, the contrast of the non-response set incongruent and the neutral trials yielded evidence for the semantic Stroop interference effect ($t_{33} = 2.53$, $p = 0.016$, $M_{\text{diff}} = 34$ ms, $d_z = 0.43$, $B_{H(0,20)} = 8.29$, $RR_{B > 3}[8, 177]$).

### 3.4.5. Crucial test 1: are the strategies effective in reducing the Stroop interference effect?

First, we examined whether or not the beneficial effect of the looking-away and blurring strategies replicated in the current experiment. We found strong evidence that both the looking-away ($t_{33} = 4.42$, $p < 0.001$, $M_{\text{diff}} = 71$ ms, $d_z = 0.76$, $B_{H(0,30)} = 297.77$, $RR_{B > 3}[7, 1.93 \times 10^4]$) and the blurring strategies ($t_{33} = 3.05$, $p = 0.005$, $M_{\text{diff}} = 43$ ms, $d_z = 0.52$, $B_{H(0,30)} = 24.93$, $RR_{B > 3}[7, 632]$) helped the participants to reduce the Stroop interference compared with the no strategy condition. Figure 3 depicts the distribution of the Stroop interference scores broken down by the strategy conditions, and table 2 presents the congruency condition means and s.d.s.

As an additional analysis, we tested whether the strategies reduced the response conflict component (incongruent RTs—non-response set RTs) of the Stroop interference effect so that our results can be compared with those of Augustinova & Ferrand [17]. The analyses revealed moderate evidence supporting that the blurring strategy reduced response conflict ($t_{33} = 1.98$, $p = 0.056$, $M_{\text{diff}} = 34$ ms, $d_z = 0.34$, $B_{H(0,30)} = 3.94$, $RR_{B > 3}[16, 64]$) and anecdotal evidence that looking-away strategy reduced response conflict ($t_{33} = 1.61$, $p = 0.117$, $M_{\text{diff}} = 29$ ms, $d_z = 0.27$, $B_{H(0,30)} = 2.40$, $RR_{1/3 < B < 3}[0, 364]$) compared with the no strategy condition. Note: these two last tests were not pre-registered.

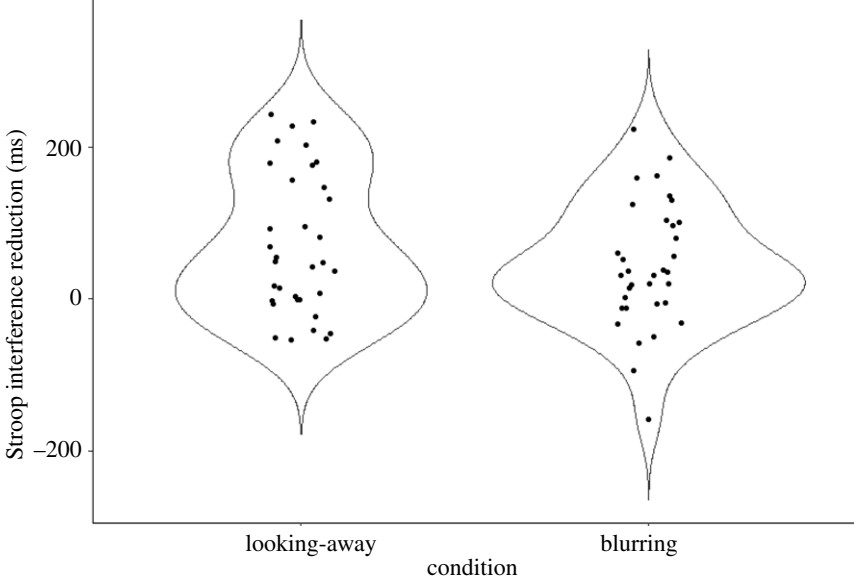

**Figure 3.** Violin plot portraying the distribution of Stroop interference score differences (ms) between the no strategy and the two strategy conditions. Each black dot represents the reduction of Stroop interference (incongruent RT—neutral RT) by a specific strategy of a single participant.

### 3.4.6. Crucial test 2: do the strategies diminish the RTs of the incongruent trials?

We found moderate evidence supporting the claim that neither the looking-away ($t_{33} = -2.35$, $p = 0.025$, $M_{diff} = -47$ ms, $d_z = -0.40$, $B_{H(0,30)} = 0.22$, $RR_{B < 1/3}[19, \infty]$) nor the blurring strategy ($t_{33} = -1.99$, $p = 0.055$, $M_{diff} = -31$ ms, $d_z = -0.34$, $B_{H(0,30)} = 0.19$, $RR_{B < 1/3}[16, \infty]$) reduced the incongruent RTs compared with the no strategy condition. In fact, we found moderate evidence regarding the slow-down of incongruent RTs for both the looking-away ($B_{H(0,30)} = 6.37$, $RR_{B > 3}[13, 175]$) and the blurring strategies ($B_{H(0,30)} = 3.98$, $RR_{B > 3}[14, 58]$).

### 3.4.7. Crucial test 3: do the strategies influence the magnitude of the semantic Stroop interference effect?

There was anecdotal evidence that the looking-away strategy reduced the semantic Stroop interference effect ($t_{33} = 2.41$, $p = 0.022$, $M_{diff} = 42$ ms, $d_z = 0.41$, $B_{H(0,10)} = 2.80$, $RR_{1/3 < B < 3}[0, 11]$). In fact, the strategy eliminated the semantic Stroop effect in the looking-away strategy condition ($t_{33} = -1.06$, $p = 0.296$, $M_{diff} = -8$ ms, $d_z = -0.18$, $B_{H(0,20)} = 0.20$, $RR_{B < 1/3}[12, \infty]$) in the case of the blurring strategy, there was no evidence either way for whether or not semantic Stroop interference was reduced ($t_{33} = 0.50$, $p = 0.617$, $M_{diff} = 8$ ms, $d_z = 0.09$, $B_{H(0,10)} = 1.07$, $RR_{1/3 < B < 3}[0, 74]$).

### 3.4.8. Supporting test of interest 1 (non-preregistered): do the strategies influence RTs in general?

We repeated the test of the main effect of strategy on the RTs of incongruent and neutral trials. The analyses revealed strong evidence for a general slow-down effect for both the looking-away ($t_{33} = 4.98$, $p < 0.001$, $M_{diff} = 83$ ms, $d_z = 0.71$, $B_{H(0,30)} = 7.66 \times 10^2$, $RR_{B > 3}[7, 2.47 \times 10^4]$) and blurring strategies ($t_{33} = 3.72$, $p < 0.001$, $M_{diff} = 53$ ms, $d_z = 0.44$, $B_{H(0,30)} = 94.59$, $RR_{B > 3}[6, 3.66 \times 10^3]$).

### 3.5. Discussion

Once more, both looking-away and blurring strategies demonstrated utility in reducing Stroop interference, and the blurring strategy approximately halved the Stroop interference effect as the word blindness suggestion tends to do when it is given to highly hypnotizable people. We also replicated the finding that neither of the strategies sped up responses during incongruent trials, and the direct comparison of the strategy conditions with the word blindness condition of a different experiment yielded evidence for their dissimilarity (for the latter analysis, see the electronic supplementary material). By introducing non-response set incongruent trials, we were able to distinguish the semantic and response conflict component of the interference effect, and we found some evidence that

the looking-away strategy alleviates both sources of conflicts, whereas for the blurring strategy, the evidence is not clear whether it solely reduces response conflict or diminishes semantic conflict as well. Importantly, we specified these two latter analyses as severe tests that can disconfirm the idea that looking away or blurring are responsible for the word blindness effect. Consequently, we ought to conclude that none of the strategies have met the criteria and are unlikely to be the strategies that highs resort to when they respond to the word blindness suggestion.

## 4. General discussion

The purpose of the project was to investigate whether cognitive or perceptual strategies can attenuate the Stroop interference effect. According to cold control theory of hypnotic responding [27], people use strategies to create the experience that was described to them in the suggestion. Hence, the investigation of strategies is crucial to assess cold control theory and to understand how highs can manage to reduce the interference effect when they respond to the word blindness suggestion. Importantly, the ability of highs to respond hypnotically (with the feeling of involuntariness) seems to be independent of their first-order executive functions, such as cognitive inhibition [48] and selective attention [50], that could help them overcome cognitive conflict during the Stroop task (see [49], for a review). We found no evidence one way or the other for a correlation between hypnotizability and the extent to which any of the strategies could decrease Stroop interference.

Next, we probed the efficiency of the four strategies: looking away, visual blurring, single-letter focus and goal-maintenance. Importantly, looking-away and blurring strategies were shown to be useful in diminishing the interference effect in both experiments, substantiating the notion that participants are able to reduce Stroop interference by consciously engaging in simple strategies; a finding that has been rarely demonstrated in the Stroop literature (cf. [19]). Nonetheless, none of these strategies should be considered as likely candidates for being the underlying mechanism of the word blindness suggestion, as they did not meet other criteria, such as reducing the RT of incongruent trials. Rather, these strategies seemed to attenuate Stroop interference by affecting the general speed of responses (see Supporting test of interest 1). Participants responded slower overall and made the RTs of different trial types more similar. This slow-down effect is not a unique finding; for instance, neutral RTs were demonstrated to increase due to experimental manipulations, such as goal-priming [16,46] or single-letter colouring and spatial cueing [64], that reduce Stroop interference. And in some cases, the latter manipulation leaves incongruent RTs unaffected or even elevates them similarly to the looking-away and blurring strategies (e.g. [65,66]). Future research is needed to understand the cognitive mechanisms underlying these processes.

The idea that goal-maintenance plays a crucial role in responding quickly and accurately to a Stroop word is well established (e.g. [9,45]) and it is embedded in many of the cognitive control models (e.g. [4,67]). It is important to note that our findings do not challenge this idea. In this project, we solely aimed to test whether a simple way to update one's goal (i.e. rehearsal of the target) is sufficient to improve performance in the Stroop task. We did not provide strong evidence one way or the other for whether highs achieve the reduction of the Stroop interference when they respond to the word blindness suggestion by internally rehearsing task instructions. However, it is still possible that the strategy with which highs reduce Stroop interference facilitates goal-maintenance. In fact, based on the finding that the word blindness suggestion operates better when the response–stimulus interval is short (500 ms) than when it is long (3500 ms), it remains possible that the strategy that highs employ influences processes related to goal-maintenance ([16]; cf. [22]).

In many cases, the word blindness suggestion impacts the RTs of neutral trials as well, and surprisingly, it reduces them (e.g. [15–17,46]). This feature of the suggestion is completely in harmony with a strategy that condenses the interference by simply speeding up all responses. However, it is unlikely that either the looking-away or the blurring strategy operates by this mechanism. First, none of these strategies reduced the neutral RTs (tables 1 and 2, and Bayesian evidence supporting that the strategies increased RTs overall). Second, we conducted a formal analysis to test this notion, in which we compared the conditions in terms of the patterns of the relationship between the general speed of responses and the magnitude of the interference effect (cf. [68,69]). These analyses confirmed that there is no relationship between the general speed of responses and the extent of the interference in the looking-away and in the blurring conditions (for the details of the analyses, see the electronic supplementary material, Exploration S3).

Finally, it is established in hypnosis research that sometimes people use different strategies to respond to the same suggestion [70,71]. Hence, one may question if none of our strategies can explain the suggestion effect, then could a combination of them, with different highs using different strategies? To assess this possibility, we assigned the participants into idiosyncratic strategy groups based on their subjective reports of strategy usage and then repeated Crucial tests 1 and 2 on these groups and on a combined dataset (For more details of the analysis see the electronic supplementary material).[7] None of the participants found the single-letter focus strategy to be the easiest to use, deeming it unlikely that this strategy contributes to the word blindness effect. While a combination of the looking-away and blurring strategies (data of Experiment 2) decreased Stroop interference, it failed to reduce incongruent RTs, deeming this combination of strategies insufficient to account for the word blindness effect. Nonetheless, the combination of the goal-maintenance, looking-away and blurring strategies (data of Experiment 1) passed Crucial test 1 and provided insensitive Bayesian evidence regarding the reduction of incongruent RTs. Thus, as it stands, the combination of these three strategies may be able to explain the word blindness suggestion, but future research is needed to settle this option.

Another possibility is that moving along the interference-overall RT slope is a strategy in itself. For example, a simple model of motivation is that it moves people along this slope, speeding up overall RT and hence reducing Stroop interference (cf. [69]). Indeed, enhanced motivation has most commonly led to an overall speeding-up of responses [13,14,72]. Nonetheless, the introduction of a reward has not often produced large reductions in Stroop effects [13,14]. More promising, setting up competition for reward in the presence of a competitive other has been shown to result in a greater than 50% reduction in Stroop interference [73].[8] One might argue that the hypnotic context provides stronger motivation for highs than monetary reward by itself or combined with competition. However, the re-analysis of an earlier study that had an identical design to the current experiment in terms of the Stroop test, but used the word blindness suggestion, revealed a raw slope of zero ($b = 0.005$ ms ms$^{-1}$, 95% CI [−0.04, 0.05]) between Stroop interference and overall RT (sum of RTs of incongruent and neutral trials) in the suggestion condition (Pilot study of [47]). That is, it does not appear that in the suggestion condition people simply move along a fixed slope, generally speeding up and thereby reducing interference. Instead, people typically reduce the RT in especially the incongruent condition when responding to the suggestion. A proper understanding of the relation of motivation to the word blindness suggestion remains to be explored.

One simple strategy still remains that was not tested in the current experiment. When highs are suggested to see meaningless words throughout the Stroop task, perhaps, they take the instructions literally, and they create the experience of meaninglessness by imagining a counterfactual world in which words are truly meaningless. One may argue that imagining a counterfactual world is not needed to create an experience of meaninglessness as subjects may simply see the words as something similar to foreign words and do not actually see them as foreign words. Nonetheless, the phenomenology of this 'seeing as' scenario does not align well with what highs generally say they experience when they are given hypnotic suggestions, such as the word blindness suggestion (e.g. [47]). Highs typically report they experience the requested phenomenology as being a genuine one (e.g. they report seeing foreign words when responding to the word blindness suggestion), which aligns better with the notion of them imagining a counterfactual world without being aware of doing so.

Seeing the Stroop words as meaningless characters by imagining a counterfactual world might influence top-down cognitive control processes in a way that helps subjects reduce Stroop interference. There are two reasons why this notion is plausible. First, imagination can have an impact on behaviour as well as on cognitive processes. For instance, mental practice can improve one's performance in golf [74]. Moreover, imagination can advance self-regulation [75], confirm, or in some cases, challenge and mitigate prejudice [76], create false autobiographical memories [77], and, finally, even enhance performance of visual search [78,79]. Second, cognitive penetrability is not completely unprecedented in the Stroop task. For instance, expectations modulated by placebo-suggestion were shown to influence performance, measured by accuracy [80], though such placebo Stroop reduction does not appear to match the word blindness suggestion in reducing Stroop interference in RTs (contrast response expectancy theory [32]). Depending on the instructions of the placebo-suggestion, it can either enhance or impair the accuracy of responses. There is, however, evidence from independent

---

[7]We thank a reviewer, Jerome Sakur, for recommending this analysis.

[8]But note in the Huguet *et al.* [73] study the baseline level of interference (and reaction times) were unusually large, resulting in reduced manual response Stroop interference values still greater than 70 ms, considerably larger than in the typical word blindness suggestion (about 35 ms).

laboratories that a prime to deteriorate one's reading abilities, by imagining what it is like to have dyslexia, can help people reduce the Stroop interference effect compared with a baseline condition with a neutral prime that has no reference to reading [59,81].

Interestingly, the dyslexia prime and word blindness suggestion phenomena share many properties. They both substantially decrease the interference effect by speeding up the RT of incongruent trials compared with no suggestion/no prime baseline conditions when the response mode is manual (see Experiment 1 of [59]; and Experiment 1 of [81]). The dyslexia prime, similarly to the word blindness suggestion, affects the response competition component of the interference while it leaves the semantic conflict component unaffected [17,59]. This latter feature of the dyslexia prime is particularly important in challenging the initially proposed mechanism, namely the de-automatization of reading account that putatively underlies these phenomena. An even more remarkable similarity between the instructions of the dyslexia prime and the word blindness suggestion experiments is that both invite participants to think about disrupting one's reading abilities. One could develop this line of thought and propose that both of these effects are achieved via deliberate strategy engagement, specifically the imagination of a counterfactual world in which words are meaningless. Theories of social priming argue that responses to primes are unintentional and purely triggered by the activation of a specific social concept [82,83]. However, there are many reasons to retain scepticism about the unintentional nature of the responses to social primes, such as the presence of demand characteristics, or the absence of valid and reliable outcome neutral tests demonstrating that the participants were not aware of the link between the social prime and the dependent variable of the experiment [84–87]. These criticisms apply to the dyslexia studies as well, deeming it plausible that the participants reduced the Stroop interference via intentional strategy usage rather than via the unintentional or automatic activation of the concept of dyslexia.

Nonetheless, the idea that imagining that one is unable to derive meaning from the Stroop words, facilitates the resolution of response competition, is a conjecture that needs to be tested. Recently, a registered report undertook such a test by requesting highs to voluntarily imagine the words during the Stroop task as meaningless characters so that they can reduce the Stroop interference compared with a baseline condition in which they are asked to not engage in imagery strategies [47]. Given the results of the current study it is likely that the subjects of the registered report did exactly what they were asked to do and used imagination rather than one of the strategies tested here to achieve the experience of meaninglessness and to reduce Stroop interference. Nevertheless, the evidence against the combination of the goal-maintenance, looking-away and blurring strategies is insensitive so the efficiency of the imagination strategy should be directly tested. Moreover, the registered report only recruited highs, so to explore the reach of the imagination strategy, it still needs to be tested whether those from the full spectrum of hypnotizability can use the imagination strategy to alleviate interference.

Finally, it is important to bear in mind that the purpose of this study and its design were inspired by the cold control theory and so the conclusions regarding the word blindness suggestion are most meaningful under the assumptions of this theory. For instance, special process theories of hypnosis, such as the integrative cognitive theory [88], the neodissociation theory [89] and the dissociated control theory of hypnosis [90,91] postulate that hypnosis influences non-metacognitive processes as well. Hence, they presume that a strategy that is unsuccessful outside of the hypnotic context may be successful when applied under hypnosis. Nonetheless, we are not aware of experimental evidence disconfirming the simpler theory, cold control, which provides the basis of the current study (see [47] for a review of the evidence in support of the core assumption of cold control theory). Moreover, the above-cited registered report deems it unlikely that in the case of the word blindness suggestion, highs would be using a strategy under hypnosis that they cannot use outside of hypnosis.

One might ask how cold control theory accounts for highs responding to the word blindness suggestion by reducing the Stroop effect, but lows do not, even without a hypnotic induction (e.g. [18]). That is, if highs do not have any special attentional or control abilities (i.e. highs and lows only differ in capacity to control awareness of intentions), how do highs reduce the Stroop effect where lows do not? One hypothesis is that highs are more motivated to respond to imaginative suggestions; if lows were incentivized to engage as much as highs, they too would reduce the Stroop effect just as much by use of their imagination. This remains a hypothesis for future research to test.

In sum, reducing interference in the Stroop task via intentional means is difficult and the current study provided compelling evidence that there are at least two strategies, looking away from the target word and visual blurring, that any subject can apply. Interestingly, none of these strategies met the criteria to be considered as a potential underlying mechanism of the word blindness suggestion, and thus the modus operandi of the word blindness suggestion remains open. Although these

findings further the mystery surrounding the word blindness suggestion, we hypothesize that imagination (i.e. imagining that the Stroop words are meaningless) may be the key strategy with which subjects reset top-down cognitive processing to comply with the request of the suggestion, and lead to the reduction of the Stroop interference.

Data accessibility. The materials, the data and the analysis script of the experiments can be retrieved from https://osf.io/6a58r.
The data are provided in electronic supplementary material [92].
Authors' contributions. B.P.: conceptualization, data curation, formal analysis, investigation, methodology, project administration, software, writing—original draft, writing—review and editing; B.A.P.: conceptualization, methodology, writing—review and editing; A.F.C.: investigation, methodology, writing—review and editing; Z.D.: conceptualization, methodology, supervision, writing—review and editing.
All authors gave final approval for publication and agreed to be held accountable for the work performed therein.
Competing interests. At the time of writing, Prof. Zoltan Dienes was a Board Member of Royal Society Open Science but had no involvement in the review or assessment of the paper. All other authors declare no competing interests. The project was not supported by any grant.
Funding. We received no funding for this study.
Acknowledgements. Bence Palfi is grateful to the Dr Mortimer and Theresa Sackler Foundation which supports the Sackler Centre for Consciousness Science.

# Appendix A. Instructions in the experimental conditions of Experiment 1

## A.1. No strategy

'This time do not use any of the strategies we have instructed you in previous blocks.

We would now like you to respond to the colour of the word on the screen as quickly and as accurately as you can.'

## A.2. Looking away

'We would like you to focus on the top-right corner of the screen throughout the following experimental block and use only your peripheral vision to identify the colour of the words that appear on the screen.

You can practice this strategy now on an example word.'

In this condition, the participants were told that they can focus on a spot that is closer to the word if they found the top-right corner to be too far away to easily identify the colour of the word.

## A.3. Blurring

'We would like you to blur your vision throughout the following experimental block by focusing on the screen as if you were looking into the distance.

You can practice this strategy now on an example word.'

## A.4. Single-letter focus

'We would like you to attend to a portion of the last coloured letter of each word in the next experimental block.

You can practice this strategy now on an example word.'

## A.5. Goal-maintenance

'We would like you to internally repeat the phrase 'displayed colour' whenever you see the fixation cross.

Please repeat the phrase until the target appears on the screen.'

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
