## [Peer Review File · Royal Society Open Science]

Review History

RSOS-202136.R0 (Original submission)

Review form: Reviewer 1

Is the manuscript scientifically sound in its present form?

Yes

Are the interpretations and conclusions justified by the results?

No

Is the language acceptable?

Yes

Do you have any ethical concerns with this paper?

No

Have you any concerns about statistical analyses in this paper?

No

Recommendation?

Reject

Comments to the Author(s)

SUMMARY. Previous studies have reported that hypnotic word blindness suggestion (that words are meaningless symbols) can reduce the size of Stroop interference effect in highly hypnotizable individuals. The authors wished to investigate how highly hypnotisable individuals manage to do this. To this end, instructions were given for four candidate strategies outside of the hypnotic context to 57 participants, screened for hypnotizability, in a (manual) Stroop task (Experiment 1). However, the pattern of results produced by these strategies did not match those of the word blindness suggestion. Experiment 2 followed up two of the strategies that reduced the Stroop interference effect with two types of incongruent distractors (color words that were the names of response colors and words that weren't). The authors conclude that "Although the current results did not unravel the mystery of the word blindness suggestion, they showed that there are multiple voluntary ways through which participants can dramatically reduce Stroop interference".

EVALUATION. This paper is generally written well (though see my comments below), and the study is well-motivated. The results indicating that the "looking away" and the "blurring" strategies reduced the Stroop interference effect, but that the pattern produced by these strategies – the general slowdown in all conditions - did not match that produced by hypnotic word blindness suggestion would be of interest to researchers interested in the Stroop task, in particular, those interested in whether the interference effect can be modulated. However, ultimately, the present study did not meet the goal set by the researchers themselves (to find out how highly hypnotizable participants manage to reduce the Stroop interference effect in response to the word blindness suggestion). Moreover, surprisingly, the authors do not report whether this effect was replicated in the highly hypnotizable participants in the present study. Some of the theoretical points are not explained very well. In particular, the authors motivate Experiment 2 to test the prediction that the strategy(ies) underlying the word blindness suggestion "should alleviate response conflict rather than semantic conflict" (p.20, bottom). However, neither what the authors mean by response competition vs. semantic conflict, nor the evidence for this claim is not explained at all. (I elaborate on this point under Specific points below). There are also minor typographical errors that suggest it was written by a student author, and the editing was not quite thorough (please see below, under Minor/specific points"). All things considered, I felt that the paper did not make a theoretical advance, and that the paper was not quite ready for publication.

Specific points

- Typos. There are a couple of places where a word was struck out and the strike-out mark remained: p.21, "how" is struck out and replaced by "whether": p.31, "good" is struck out and replaced by "strong"
- Typos. P.26, end of first para, where it says "Table 2", I'm reasonably sure the authors mean Table 4.
- p.20 bottom, "Another key characteristic of the word blindness suggestion is that it seems to reduce interference by attenuating response competition and not by de-automatising reading per se". This claim is just stated, and a bunch of references listed "(Augustinova & Ferrand, 2012; Palfi Parris, Seth, & Dienes, 2018; Parris, Dienes & Hodgson, 2013; contrast Raz, Fan & Posner, 2005)", without describing the evidence presented by these studies. I am familiar with the Augustinova & Ferrand (2012) study in which the hypnotic suggestion of wordblindness reduced the standard Stroop interference effect with response-incongruent words but not the interference produced by color-associated words (e.g., "LEMON" presented in blue). But as the present authors point out later in Footnote 3 (p.22), the interference effect produced by the color-associated words is small and not robust (Kinoshita, Mills & Norris, 2018), so this result may just indicate a lack of sensitivity to the manipulation, rather than evidence that the wordblindness suggestion reduces response competition, but not semantic conflict. I am not familiar with the

other studies by the authors' group (Palfi, et al., 2018 – the References indicates this is a ms in preparation; Parris et al., 2013), and in any case the evidence needs to be described here to provide the rationale for testing this claim in Experiment 2. This is particularly important as it appears that the claim that the hypnotic suggestion works by reducing response competition does not have universal support (“contrast Raz, Fan & Posner, 2005”).

- I note that some of the Stroop results reported in the literature as being immune to the hypnotic suggestion of wordblindness are not factually correct. For example, Raz & Campbell (2011) reported that while the hypnotic suggestion reduced the overall Stroop interference effect it did not affect the negative priming effect. We (Mills, Kinoshita & Norris, 2019, *Frontiers in Psychology*, “No negative priming effect in the manual Stroop task”) argued that this claim likely reflects the incorrect way in which the negative priming effect is calculated in the study (confounded with the response repetition effect), as the manual Stroop task generally does not produce the negative priming effect in the first place. So I think it is really important to replicate the pattern of Stroop results supposedly produced by the hypnotic suggestion of wordblindness reported in the literature before investigating what mechanism explains the pattern of results.

- Contingency. Recent Stroop studies have pointed out that the congruent condition (e.g., the word RED presented in red colour) and the incongruent condition (e.g., the word RED presented in green colour) differ in the word-colour contingency when there are more than 2 response colours and the number of congruent trials and incongruent trials are equated, as was done here, unless only one incongruent colour was used with an incongruent word. It is not clear from the description if this is what was done here in Experiment 1. Please make this clear. Contingency has a large effect on Stroop interference (response to color is much faster for high-contingency items (e.g., Lin & MacLeod, 2017, *JEP:LMC*).

Review form: Reviewer 2

Is the manuscript scientifically sound in its present form?

Yes

Are the interpretations and conclusions justified by the results?

Yes

Is the language acceptable?

Yes

Do you have any ethical concerns with this paper?

No

Have you any concerns about statistical analyses in this paper?

Yes

Recommendation?

Accept with minor revision (please list in comments)

Comments to the Author(s)

The manuscript reports two experiments that examine the mechanism(s) by which the word-blindness effect arises during hypnotic suggestion studies of the Stroop Task. The experiments are well thought out and contribute new insights into word-blindness effects in the Stroop task. That said, the introduction to the manuscript juggles two concurrent issues (1) by what

mechanism/strategy do the word-blindness effects arise and (2) are these related to hypnotisability. Overall, the paper is well written and I think it makes a sufficient empirical contribution to warrant publication. I do, however, have some concerns about the structure of the manuscript recommendations for some revisions to the manuscript. These concerns primarily center on the inconsistent weighting of the two goals throughout the manuscript, and explicit statements to this effect.

General Comments

The introduction establishes two goals for the manuscript: (1) by what “strategy” does the word-blindness effect arise and (2) is the strategy related to hypnotisability. The first of these goals is clear and easy to follow; the latter not so much. Perhaps the authors could re-organize the introduction so that all of the discussion with regard to strategies is dealt with first and then delve into the second goal? That said, I’m not sure the experiment can even address the theoretical motivation for latter goal. For instance, is there an outcome that would actually falsify the account?

According to the authors, there are two (maybe three) signature effects that need to be observed by a strategy if it is going to provide a reasonable explanation of behaviour under word blindness conditions with highly suggestible participants. The first signature is a reduction in Stroop Interference. The second signature is a speed up on interference trials (as opposed to a slowdown across neutral and congruent trials). A possible third signature, which the authors admit is more variable is the absence of a reduction in Stroop facilitation. The situation surrounding the latter should probably be unpacked a little more (an additional sentence would be enough and I’m not suggesting that the authors need to test this). My concern is that the hypotheses/ predictions of Experiment 1 do not reflect the earlier purported importance of these two effects. In addition, the second signature is not described as a critical test in the results of Experiment 1. This stands in contrast to the analyses of Experiment 2, where they are both clearly emphasized. And again the conjunction of these two effects are focus of the General Discussion.

The flow of the manuscript with regard to the other goal of the paper (testing a (possible???) prediction of cold-control theory). The authors start with this issue, leave to discuss strategies, then return to the theory of cold-control theory towards the end of introduction (the motivation for goal 2, which is abandoned after Experiment 1 and really doesn’t play any necessary role in the manuscript).

In the general discussion, I’m not sure where the statement that “these strategies seemed to attenuate Stroop interference by affecting the general speed of responses” comes from. The authors quickly move on to clarify what the mean, but if you are going to do this, why include a description of a main effect that is never tested for and clearly doesn’t explain the data. I recommend that the authors discuss a few more findings from the Stroop literature. In part because I think it would be worthwhile for the authors to articulate that the word blindness effect could be mislabelled, if it is thought of as an explanation as opposed to the instructions given to participants. In my mind, one issue is whether the strategies examined in the manuscript can explain the word blindness effect (they do not), whereas another issue is whether the word blindness induction is affecting reading at all. For instance, reducing readability should affect both facilitation and interference, moving both towards the neutral condition – especially if the neutral condition is not a word. With this in mind, a paper by Melara and Mounts (1993) on dimensional discriminability may be relevant. Further, it may be worthwhile discussing factors that affect both facilitation and interference, such as the response set size (I think MacLeod, 1991 might provide a discussion of this), and factors that affect only interference such as the proportion of congruent/incongruent trials, which Lowe and Mitterer (1982) suggested specifically targets response conflict resolution.

The authors might also consider discussing a paper by Reynolds, Kwan & Smilek (2010) when discussing factors that modulate the Stroop effect at the very beginning of the manuscript. This is an oft neglected paper that goes some way towards examining the bottom up processes that modulate the Stroop effect and might provide a useful framework for thinking about single letter colouring and how the word blindness could affect processing. However, I think this might be better served in the introduction.

One other element from the Stroop literature that should be elaborated on is the issue of a “neutral” condition. In particular, I feel that the discussion of neutral RTs in the General Discussion cannot occur without a discussion of the problems with the concept of a neutral condition. In particular the choice to use neutral words instead of over possible baselines.

In the methods: What is the size of the screen? The look away location could make it difficult to perceive the color. For instance, a 20 inch screen would mean that they were looking approximately 25 degrees from the colour. I realize that participants were told that they didn't have to look in the corner if it was too far, but still this info is relevant.

I understand the value of replicating the methods reported by Raz et al. (2002) however, some deviations would have been nice. For instance, if the hypnotic suggestion effect is largest at short RSIs why not include or use that condition? See also p31 of the discussion.

More troubling was their decision to not label the response keys (Raz et al., 2002 did label them). The lack of substantial practice (36 trials vs. 32 in Raz et al.) means that the participants were probably still learning the key-mappings well into the first block. This has implications for their analysis of the block 1 data.

The analyses of “carry-over” effects, in which the focused solely on the data from Block 1, is not really an analysis of carry-over effects. Instead, it is an analysis without carry-over effects. I realize that this is a minor detail, but it is worth pointing out. Further, this given the lack of practice the interpretability of the first block problematic. For instance, if participants are still learning the response mapping, this could interfere differently with each of strategy manipulations. Further, the authors do not report what the sample size is per condition – strategy was randomly assigned to block. Finally, this analysis is a between subjects comparison. Therefore, the power to detect any effect would be weak at the best of times, let alone with all of these problems. Although there is no formal analysis of this data in the manuscript the authors do draw conclusions from the data, which are presented in a table. I recommend that the authors remove the analysis or tone down their conclusions to adequately reflect the interpretability problems.

Analyses

I should note that I did not focus on the Bayesian analyses in too much detail. That said, I did note that it does not appear that the authors dealt with how observing an effect in the opposite direction affects Bayesian calculations.

There are also some missing effect size values in the analyses reported on page 25 of the manuscript.

I am blown away by the size of the Stroop effects in the present study. They are absolutely huge given the methodology.

I am not a huge fan of violin plots although I recognize that they are all the rage lately. My concern is that they are largely useless in within-subjects designs as the correlation between

conditions is critical to their interpretation. This is also true of the standard deviations in the tables and confidence intervals in graphs – see Masson & Loftus.

Minor comments

p4, ln 35. The word to should be removed.

p4, ln 50. I'm not sure if it is worth mentioning, but the stroop effect is robust at a group level but not at the level of individuals.

p6, ln 8. the Bowers (1990) reference is missing

Review form: Reviewer 3 (Jérôme Sackur)

Is the manuscript scientifically sound in its present form?

Yes

Are the interpretations and conclusions justified by the results?

Yes

Is the language acceptable?

Yes

Do you have any ethical concerns with this paper?

No

Have you any concerns about statistical analyses in this paper?

No

Recommendation?

Accept with minor revision (please list in comments)

Comments to the Author(s)

A "word blindness" suggestion can decrease the Stroop interference effect, when given to highly hypnotisable participants. The paper argues that, within the context of the cold control theory (CCT) of hypnosis (according to which hypnosis removes metacognitive awareness of the strategy at work), four plausible strategies could be employed. It reports that two of them "looking away" and "blurring" are efficient, but that they do not exactly reproduce the pattern of results obtained with the word blindness suggestion. Hence the author conclude that the strategy employed by participants under a word blindness suggestion has yet to be found.

The paper is clear and provides a useful contribution to the current debates about the mechanisms of hypnosis. I have a few general remarks:

1/ At times, the authors might seem to forget that theirs is a test of the conjunction of the CCT and of some particular strategies. It might be for instance that the CCT is not the complete story, and that under hypnosis Highs do things a bit differently than outside hypnosis. For instance, It might be that under hypnosis, Highs "look away" in a specific manner, which they are not able to reproduce when given the explicit instruction to look away without hypnosis. Thus the inadequacy of the strategies is most meaningful under the assumption of the CCT (and there are many reasons to accept this theory, but the present paper is not one of them, so the assumptive nature of the CCT should not be hidden or forgotten).

2/ I think the discussion about what it would mean to take the word blindness suggestion literally should not be deferred to the Discussion. Otherwise, it seems that the authors take it for granted that the suggestion is necessarily metaphorical. On p. 4, the authors seem to take it as self-evident that a word blindness suggestion of the form "you will see the words as gibberish" is not directly actionable by participants. (As if you would say: "levitate and touch the ceiling", and that some participants implement that with the "grab a chair" strategy, some with the "jump and extend your arms" strategy, etc...). One might argue that it is intrinsic to hypnotic suggestions to be obscure, ambiguous, metaphorical, etc. But it is not necessarily so, and if this is the authors' stance, they should argue for it. I think that at least, the authors need to present some argument in the Introduction as to why they don't test the literal content of the suggestion as a strategy.

Furthermore, when the authors do address this issue in the Discussion, they interpret it as the imagination of a counterfactual world (ie, participants would imagine a world in which they have dyslexia). But there might be an even more literal interpretation, along the lines of the concepts of "aspect switching" and "seeing as" as they are discussed in aesthetics and philosophy (cf notably Wittgenstein, /Philosophical Investigations/, /Remarks on the philosophy of psychology/). To see a particular cloud as the head of a cat, one does not need to imagine a counterfactual world in which bodyless giant cats fly.

3/ The systematicity of the statistical analysis is of course a good thing, but it sometimes makes the results hard to follow. Below I detail some of my difficulties.

* Introduction

p. 6 l. 52 When referring to attention, here I would prefix it with "spatial". Because, obviously, if any strategy is to be efficacious, it is a form of attentional strategy, although not necessarily spatial.

p. 9 I'm not sure the last short paragraph of the Introduction is really necessary.

* Stimulus and apparatus

No detail is provided concerning the psychophysical properties of the colors. Values in the RGB space are not very informative, but it would be nice to have at least subjective judgments. In my experience, with "out of the box" software colored stimuli, yellows make for really less legible words. (Acknowledgely, it is orthogonal to the hypotheses, but still...) This is perhaps more important in Exp. 2, where two color sets are employed.

p. 12 Statistical Analyses

I understand the logic of analyzing difference scores, but I think raw RTs would still be informative, and indeed you report them in table 1. You do have rather large between condition differences in mean speed. I think they are interesting as such. Indeed one of the leitmotiv of the analyses is whether such and such strategy "reduces the RTs of incongruent trials" (cf p. 16 l34 for instance). Accordingly, sometimes the difference scores are computed across congruent and incongruent trials within instruction condition, and compared with the analogous difference within the no instruction condition; sometimes they are computed between conditions within the incongruent trials. Essentially, if I understand well, this could all be summed up in an ANOVA model with two main effects (condition and congruency) and an interaction of condition and congruency. To me such a presentation of the results would be much clearer, followed by the Bayesian t-tests for the contrasts of interest.

p. 13 l23: I don't understand the reference of the "60ms and 105ms, respectively". From the next sentence, I infer that 60ms is the prior for the Stroop interference, but what is 105ms?

* Design and procedure

The time sequence of the trial is a not really precise. Am I right to infer that the stimulus has a fixed duration of 2 s. yet the inter-stimulus interval is variable?

p. 12 / Appendix: The strategies must have been explained at first before the experiment started? Otherwise, what were participants to make of the "no-strategy" instruction "This time do not use any of the strategies we have instructed you in previous blocks." in case they happened to start with a "no-strategy" block?

p. 12 / Appendix Why is it that the "No-strategy" instruction is the only one to have a speed and accuracy instruction ("as quickly and as accurately as you can")? I don't expect that to have a huge impact, but still, one could argue that this creates a self-evaluation pressure (and indeed, the no-strategy condition is the fastest of all.)

* Results

I feel that an error rate of 8.2% is not totally negligible, on account of the slow pace of the experiment. I think it might be valuable to test whether there are congruency effects in the error rates, and whether they are modulated by the strategies.

p. 14 l44 "The error rates were low..." it must be the proportion of outliers, not the errors?

p. 15 "Crucial test 1" This paragraph demands that the reader maintain the information that the "Stroop interference effect" (resp. the "Stroop effect") is the difference between incongruent and neutral trials (resp. incongruent and congruent).

p. 19 The legend of Table 2 is not specific enough. I think it would be important to know how many participants are included in each row of the table. If I understand well, that number would be around 12 (57/5)?

* Experiment 2

** Results

Same remarks as above about the error rate and outlier proportions.

p. 25 With respect to the comparison of the response sets A and B, I would guess that the colors of the original one are much more saturated and luminous, which might explain why there is a small difference between the two sets? I surmise that equiluminant colors would help in reducing the difference.

* General Discussion

It would be interesting to discuss the ease of each strategy (as reported subjectively by participants) with respect to the cold control theory of hypnosis. It may be that this value is correlated with SWASH score...

The authors do not envision the possibility that each individual participant may have a preferred (most efficient for them) strategy. So it could be that the set of the four strategies compared in the paper is perfectly adequate and sufficient to explain the impact of the word blindness suggestion, but that you would need to pick the right one for each participant--something Highs might be

able to do spontaneously under hypnosis. This would not be totally surprising, considering how hypnosis is used in clinical settings. Note that it would not be hard to test that with the present data set: select the best strategy for each participant (the one that reduces the Stroop effect the most and speeds up incongruent trials), and redo the analysis on this idiosyncratic strategy. It would also be interesting to check whether the strategy thus selected is the one that is subjectively easier to apply. (Indeed, one may proceed based on this subjective judgment: select the idiosyncratic strategy based on the subjective judgment at the end of each block).

At times, it seems that the authors tend to treat the cold-control theory of hypnosis as a fact. For instance, p. 30: "Nonetheless, none of these strategies should be considered as likely candidates for being the underlying mechanism of the word blindness suggestion, as they did not meet other criteria, such as reducing the RT of incongruent trials." This conclusion holds under the cold control theory, but not necessarily otherwise: suppose for instance that highs after hypnotic induction do modify their first order processes, and not simply their metacognitive access to them (contra Cold-Control). Then, one could imagine that in this state they discover how to blur their vision in a much more efficient way than without hypnosis (in addition to doing that involuntarily). We would then have the same strategy (blurring), but performed at various levels of efficiencies.

p. 35 l. 39 I don't see the motivation for the reservation here expressed: "it still needs to be explored whether those from the full spectrum of hypnotisability can use this strategy to alleviate the interference, and whether the influence of imagination may generalise to other cognitive tasks, such as the flanker (Eriksen & Eriksen, 1974) or Simon task (Simon & Wolf, 1963)". The first part might be needed under the assumption of the cold control theory, but again, if one relaxes the constrain of cold control, it might simply be the case that Highs are able to achieve this feat of imagination, while Lows and Mediums would not. As for the last part, it seems not quite relevant to the precise discussion about the Stroop case.

p. 4 l. 18 "the by the"

p. 6 l. 34 McLeod

p. 16 ll 12- 15 trailing zeros could be removed.

p. 26 l. 30 "Table 2" -> 3?

p. 27 Figure 3 caption: the "Stroop interference scores" are computed with respect to both incongruent conditions, or only for the traditional semantic + response conflict condition?

I always sign my reviews

Jérôme Sackur

Decision letter (RSOS-202136.R0)

Dear Dr Palfi

The Editors assigned to your paper RSOS-202136 "Strategies that reduce Stroop interference" have now received comments from reviewers and would like you to revise the paper in accordance with the reviewer comments and any comments from the Editors. Please note this decision does not guarantee eventual acceptance.

Please submit your revised manuscript and required files (see below) no later than 21 days from today's (ie 04-Jun-2021) date. Note: the ScholarOne system will 'lock' if submission of the revision is attempted 21 or more days after the deadline. If you do not think you will be able to meet this deadline please contact the editorial office immediately.

on behalf of Dr Denes Szucs (Associate Editor) and Essi Viding (Subject Editor)
openscience@royalsociety.org

Reviewer comments to Author:

Reviewer: 1

Comments to the Author(s)

SUMMARY. Previous studies have reported that hypnotic word blindness suggestion (that words are meaningless symbols) can reduce the size of Stroop interference effect in highly hypnotizable individuals. The authors wished to investigate how highly hypnotisable individuals manage to do this. To this end, instructions were given for four candidate strategies outside of the hypnotic context to 57 participants, screened for hypnotizability, in a (manual) Stroop task (Experiment 1). However, the pattern of results produced by these strategies did not match those of the word blindness suggestion. Experiment 2 followed up two of the strategies that reduced the Stroop interference effect with two types of incongruent distractors (color words that were the names of response colors and words that weren't). The authors conclude that "Although the current results did not unravel the mystery of the word blindness suggestion, they

showed that there are multiple voluntary ways through which participants can dramatically reduce Stroop interference”.

EVALUATION. This paper is generally written well (though see my comments below), and the study is well-motivated. The results indicating that the “looking away” and the “blurring” strategies reduced the Stroop interference effect, but that the pattern produced by these strategies – the general slowdown in all conditions – did not match that produced by hypnotic word blindness suggestion would be of interest to researchers interested in the Stroop task, in particular, those interested in whether the interference effect can be modulated. However, ultimately, the present study did not meet the goal set by the researchers themselves (to find out how highly hypnotizable participants manage to reduce the Stroop interference effect in response to the word blindness suggestion). Moreover, surprisingly, the authors do not report whether this effect was replicated in the highly hypnotizable participants in the present study. Some of the theoretical points are not explained very well. In particular, the authors motivate Experiment 2 to test the prediction that the strategy(ies) underlying the word blindness suggestion “should alleviate response conflict rather than semantic conflict” (p.20, bottom). However, neither what the authors mean by response competition vs. semantic conflict, nor the evidence for this claim is not explained at all. (I elaborate on this point under Specific points below). There are also minor typographical errors that suggest it was written by a student author, and the editing was not quite thorough (please see below, under Minor/specific points”). All things considered, I felt that the paper did not make a theoretical advance, and that the paper was not quite ready for publication.

Specific points

- Typos. There are a couple of places where a word was struck out and the strike-out mark remained: p.21, “how” is struck out and replaced by “whether”: p.31, “good” is struck out and replaced by “strong”
- Typos. P.26, end of first para, where it says “Table 2”, I’m reasonably sure the authors mean Table 4.
- p.20 bottom, “Another key characteristic of the word blindness suggestion is that it seems to reduce interference by attenuating response competition and not by de-automatising reading per se”. This claim is just stated, and a bunch of references listed “(Augustinova & Ferrand, 2012; Palfi Parris, Seth, & Dienes, 2018; Parris, Dienes & Hodgson, 2013; contrast Raz, Fan & Posner, 2005)”, without describing the evidence presented by these studies. I am familiar with the Augustinova & Ferrand (2012) study in which the hypnotic suggestion of wordblindness reduced the standard Stroop interference effect with response-incongruent words but not the interference produced by color-associated words (e.g., “LEMON” presented in blue). But as the present authors point out later in Footnote 3 (p.22), the interference effect produced by the color-associated words is small and not robust (Kinoshita, Mills & Norris, 2018), so this result may just indicate a lack of sensitivity to the manipulation, rather than evidence that the wordblindness suggestion reduces response competition, but not semantic conflict. I am not familiar with the other studies by the authors’ group (Palfi, et al., 2018 – the References indicates this is a ms in preparation; Parris et al., 2013), and in any case the evidence needs to be described here to provide the rationale for testing this claim in Experiment 2. This is particularly important as it appears that the claim that the hypnotic suggestion works by reducing response competition does not have universal support (“contrast Raz, Fan & Posner, 2005”).
- I note that some of the Stroop results reported in the literature as being immune to the hypnotic suggestion of wordblindness are not factually correct. For example, Raz & Campbell (2011) reported that while the hypnotic suggestion reduced the overall Stroop interference effect it did not affect the negative priming effect. We (Mills, Kinoshita & Norris, 2019, *Frontiers in Psychology*, “No negative priming effect in the manual Stroop task”) argued that this claim likely reflects the incorrect way in which the negative priming effect is calculated in the study

(confounded with the response repetition effect), as the manual Stroop task generally does not produce the negative priming effect in the first place. So I think it is really important to replicate the pattern of Stroop results supposedly produced by the hypnotic suggestion of wordblindness reported in the literature before investigating what mechanism explains the pattern of results.

- Contingency. Recent Stroop studies have pointed out that the congruent condition (e.g., the word RED presented in red colour) and the incongruent condition (e.g., the word RED presented in green colour) differ in the word-colour contingency when there are more than 2 response colours and the number of congruent trials and incongruent trials are equated, as was done here, unless only one incongruent colour was used with an incongruent word. It is not clear from the description if this is what was done here in Experiment 1. Please make this clear. Contingency has a large effect on Stroop interference (response to color is much faster for high-contingency items (e.g., Lin & MacLeod, 2017, JEP:LMC).

Reviewer: 2

Comments to the Author(s)

The manuscript reports two experiments that examine the mechanism(s) by which the word-blindness effect arises during hypnotic suggestion studies of the Stroop Task. The experiments are well thought out and contribute new insights into word-blindness effects in the Stroop task.

That said, the introduction to the manuscript juggles two concurrent issues (1) by what mechanism/strategy do the word-blindness effects arise and (2) are these related to hypnotisability. Overall, the paper is well written and I think it makes a sufficient empirical contribution to warrant publication. I do, however, have some concerns about the structure of the manuscript recommendations for some revisions to the manuscript. These concerns primarily center on the inconsistent weighting of the two goals throughout the manuscript, and explicit statements to this effect.

General Comments

The introduction establishes two goals for the manuscript: (1) by what “strategy” does the word-blindness effect arise and (2) is the strategy related to hypnotisability. The first of these goals is clear and easy to follow; the latter not so much. Perhaps the authors could re-organize the introduction so that all of the discussion with regard to strategies is dealt with first and then delve into the second goal? That said, I’m not sure the experiment can even address the theoretical motivation for latter goal. For instance, is there an outcome that would actually falsify the account?

According to the authors, there are two (maybe three) signature effects that need to be observed by a strategy if it is going to provide a reasonable explanation of behaviour under word blindness conditions with highly suggestible participants. The first signature is a reduction in Stroop Interference. The second signature is a speed up on interference trials (as opposed to a slowdown across neutral and congruent trials). A possible third signature, which the authors admit is more variable is the absence of a reduction in Stroop facilitation. The situation surrounding the latter should probably be unpacked a little more (an additional sentence would be enough and I’m not suggesting that the authors need to test this). My concern is that the hypotheses/ predictions of Experiment 1 do not reflect the earlier purported importance of these two effects. In addition, the second signature is not described as a critical test in the results of Experiment 1. This stands in contrast to the analyses of Experiment 2, where they are both clearly emphasized. And again the conjunction of these two effects are focus of the General Discussion.

The flow of the manuscript with regard to the other goal of the paper (testing a (possible???) prediction of cold-control theory. The authors start with this issue, leave to discuss strategies,

then return to the theory of cold-control theory towards the end of introduction (the motivation for goal 2, which is abandoned after Experiment 1 and really doesn't play any necessary role in the manuscript).

In the general discussion, I'm not sure where the statement that "these strategies seemed to attenuate Stroop interference by affecting the general speed of responses" comes from. The authors quickly move on to clarify what they mean, but if you are going to do this, why include a description of a main effect that is never tested for and clearly doesn't explain the data. I recommend that the authors discuss a few more findings from the Stroop literature. In part because I think it would be worthwhile for the authors to articulate that the word blindness effect could be mislabelled, if it is thought of as an explanation as opposed to the instructions given to participants. In my mind, one issue is whether the strategies examined in the manuscript can explain the word blindness effect (they do not), whereas another issue is whether the word blindness induction is affecting reading at all. For instance, reducing readability should affect both facilitation and interference, moving both towards the neutral condition – especially if the neutral condition is not a word. With this in mind, a paper by Melara and Mounts (1993) on dimensional discriminability may be relevant. Further, it may be worthwhile discussing factors that affect both facilitation and interference, such as the response set size (I think MacLeod, 1991 might provide a discussion of this), and factors that affect only interference such as the proportion of congruent/incongruent trials, which Lowe and Mitterer (1982) suggested specifically targets response conflict resolution.

The authors might also consider discussing a paper by Reynolds, Kwan & Smilek (2010) when discussing factors that modulate the Stroop effect at the very beginning of the manuscript. This is an oft neglected paper that goes some way towards examining the bottom up processes that modulate the Stroop effect and might provide a useful framework for thinking about single letter colouring and how the word blindness could affect processing. However, I think this might be better served in the introduction.

One other element from the Stroop literature that should be elaborated on is the issue of a "neutral" condition. In particular, I feel that the discussion of neutral RTs in the General Discussion cannot occur without a discussion of the problems with the concept of a neutral condition. In particular the choice to use neutral words instead of over possible baselines.

In the methods: What is the size of the screen? The look away location could make it difficult to perceive the color. For instance, a 20 inch screen would mean that they were looking approximately 25 degrees from the colour. I realize that participants were told that they didn't have to look in the corner if it was too far, but still this info is relevant.

I understand the value of replicating the methods reported by Raz et al. (2002) however, some deviations would have been nice. For instance, if the hypnotic suggestion effect is largest at short RSIs why not include or use that condition? See also p31 of the discussion.

More troubling was their decision to not label the response keys (Raz et al., 2002 did label them). The lack of substantial practice (36 trials vs. 32 in Raz et al.) means that the participants were probably still learning the key-mappings well into the first block. This has implications for their analysis of the block 1 data.

The analyses of "carry-over" effects, in which they focused solely on the data from Block 1, is not really an analysis of carry-over effects. Instead, it is an analysis without carry-over effects. I realize that this is a minor detail, but it is worth pointing out. Further, this given the lack of practice the interpretability of the first block is problematic. For instance, if participants are still learning the response mapping, this could interfere differently with each of strategy

manipulations. Further, the authors do not report what the sample size is per condition – strategy was randomly assigned to block. Finally, this analysis is a between subjects comparison. Therefore, the power to detect any effect would be weak at the best of times, let alone with all of these problems. Although there is no formal analysis of this data in the manuscript the authors do draw conclusions from the data, which are presented in a table. I recommend that the authors remove the analysis or tone down their conclusions to adequately reflect the interpretability problems.

Analyses

I should note that I did not focus on the Bayesian analyses in too much detail. That said, I did note that it does not appear that the authors dealt with how observing an effect in the opposite direction affects Bayesian calculations.

There are also some missing effect size values in the analyses reported on page 25 of the manuscript.

I am blown away by the size of the Stroop effects in the present study. They are absolutely huge given the methodology.

I am not a huge fan of violin plots although I recognize that they are all the rage lately. My concern is that they are largely useless in within-subjects designs as the correlation between conditions is critical to their interpretation. This is also true of the standard deviations in the tables and confidence intervals in graphs – see Masson & Loftus.

Minor comments

p4, ln 35. The word to should be removed.

p4, ln 50. I'm not sure if it is worth mentioning, but the stroop effect is robust at a group level but not at the level of individuals.

p6, ln 8. the Bowers (1990) reference is missing

Reviewer: 3

Comments to the Author(s)

A "word blindness" suggestion can decrease the Stroop interference effect, when given to highly hypnotisable participants. The paper argues that, within the context of the cold control theory (CCT) of hypnosis (according to which hypnosis removes metacognitive awareness of the strategy at work), four plausible strategies could be employed. It reports that two of them "looking away" and "blurring" are efficient, but that they do not exactly reproduce the pattern of results obtained with the word blindness suggestion. Hence the author conclude that the strategy employed by participants under a word blindness suggestion has yet to be found.

The paper is clear and provides a useful contribution to the current debates about the mechanisms of hypnosis. I have a few general remarks:

1/ At times, the authors might seem to forget that theirs is a test of the conjunction of the CCT and of some particular strategies. It might be for instance that the CCT is not the complete story, and that under hypnosis Higs do things a bit differently than outside hypnosis. For instance, It might be that under hypnosis, Higs "look away" in a specific manner, which they are not able to reproduce when given the explicit instruction to look away without hypnosis. Thus the inadequacy of the strategies is most meaningful under the assumption of the CCT (and there are many reasons to accept this theory, but the present paper is not one of them, so the assumptive nature of the CCT should not be hidden or forgotten).

2/ I think the discussion about what it would mean to take the word blindness suggestion literally should not be deferred to the Discussion. Otherwise, it seems that the authors take it for granted that the suggestion is necessarily metaphorical. On p. 4, the authors seem to take it as self-evident that a word blindness suggestion of the form "you will see the words as gibberish" is not directly actionable by participants. (As if you would say: "levitate and touch the ceiling", and that some participants implement that with the "grab a chair" strategy, some with the "jump and extend your arms" strategy, etc...). One might argue that it is intrinsic to hypnotic suggestions to be obscure, ambiguous, metaphorical, etc. But it is not necessarily so, and if this is the authors' stance, they should argue for it. I think that at least, the authors need to present some argument in the Introduction as to why they don't test the literal content of the suggestion as a strategy.

Furthermore, when the authors do address this issue in the Discussion, they interpret it as the imagination of a counterfactual world (ie, participants would imagine a world in which they have dyslexia). But there might be an even more literal interpretation, along the lines of the concepts of "aspect switching" and "seeing as" as they are discussed in aesthetics and philosophy (cf notably Wittgenstein, /Philosophical Investigations/, /Remarks on the philosophy of psychology/). To see a particular cloud as the head of a cat, one does not need to imagine a counterfactual world in which bodyless giant cats fly.

3/ The systematicity of the statistical analysis is of course a good thing, but it sometimes makes the results hard to follow. Below I detail some of my difficulties.

* Introduction

p. 6 l. 52 When referring to attention, here I would prefix it with "spatial". Because, obviously, if any strategy is to be efficacious, it is a form of attentional strategy, although not necessarily spatial.

p. 9 I'm not sure the last short paragraph of the Introduction is really necessary.

* Stimulus and apparatus

No detail is provided concerning the psychophysical properties of the colors. Values in the RGB space are not very informative, but it would be nice to have at least subjective judgments. In my experience, with "out of the box" software colored stimuli, yellows make for really less legible words. (Acknowledgely, it is orthogonal to the hypotheses, but still...) This is perhaps more important in Exp. 2, where two color sets are employed.

p. 12 Statistical Analyses

I understand the logic of analyzing difference scores, but I think raw RTs would still be informative, and indeed you report them in table 1. You do have rather large between condition differences in mean speed.

I think they are interesting as such. Indeed one of the leitmotiv of the analyses is whether such and such strategy "reduces the RTs of incongruent trials" (cf p. 16 l34 for instance). Accordingly, sometimes the difference scores are computed across congruent and incongruent trials within instruction condition, and compared with the analogous difference within the no instruction condition; sometimes they are computed between conditions within the incongruent trials. Essentially, if I understand well, this could all be summed up in an ANOVA model with two main effects (condition and congruency) and an interaction of condition and congruency. To me such a presentation of the results would be much clearer, followed by the Bayesian t-tests for the contrasts of interest.

p. 13 l23: I don't understand the reference of the "60ms and 105ms, respectively". From the next sentence, I infer that 60ms is the prior for the Stroop interference, but what is 105ms?

* Design and procedure

The time sequence of the trial is a not really precise. Am I right to infer that the stimulus has a fixed duration of 2 s. yet the inter-stimulus interval is variable?

p. 12 / Appendix: The strategies must have been explained at first before the experiment started? Otherwise, what were participants to make of the "no-strategy" instruction "This time do not use any of the strategies we have instructed you in previous blocks." in case they happened to start with a "no-strategy" block?

p. 12 / Appendix Why is it that the "No-strategy" instruction is the only one to have a speed and accuracy instruction ("as quickly and as accurately as you can")? I don't expect that to have a huge impact, but still, one could argue that this creates a self-evaluation pressure (and indeed, the no-strategy condition is the fastest of all.)

* Results

I feel that an error rate of 8.2% is not totally negligible, on account of the slow pace of the experiment. I think it might be valuable to test whether there are congruency effects in the error rates, and whether they are modulated by the strategies.

p. 14 l44 "The error rates were low..." it must be the proportion of outliers, not the errors?

p. 15 "Crucial test 1" This paragraph demands that the reader maintain the information that the "Stroop interference effect" (resp. the "Stroop effect") is the difference between incongruent and neutral trials (resp. incongruent and congruent).

p. 19 The legend of Table 2 is not specific enough. I think it would be important to know how many participants are included in each row of the table. If I understand well, that number would be around 12 (57/5)?

* Experiment 2

** Results

Same remarks as above about the error rate and outlier proportions.

p. 25 With respect to the comparison of the response sets A and B, I would guess that the colors of the original one are much more saturated and luminous, which might explain why there is a small difference between the two sets? I surmise that equiluminant colors would help in reducing the difference.

* General Discussion

It would be interesting to discuss the ease of each strategy (as reported subjectively by participants) with respect to the cold control theory of hypnosis. It may be that this value is correlated with SWASH score...

The authors do not envision the possibility that each individual participant may have a preferred (most efficient for them) strategy. So it could be that the set of the four strategies compared in the paper is perfectly adequate and sufficient to explain the impact of the word blindness suggestion,

but that you would need to pick the right one for each participant--something Highs might be able to do spontaneously under hypnosis. This would not be totally surprising, considering how hypnosis is used in clinical settings. Note that it would not be hard to test that with the present data set: select the best strategy for each participant (the one that reduces the Stroop effect the most and speeds up incongruent trials), and redo the analysis on this idiosyncratic strategy. It would also be interesting to check whether the strategy thus selected is the one that is subjectively easier to apply. (Indeed, one may proceed based on this subjective judgment: select the idiosyncratic strategy based on the subjective judgment at the end of each block).

At times, it seems that the authors tend to treat the cold-control theory of hypnosis as a fact. For instance, p. 30: "Nonetheless, none of these strategies should be considered as likely candidates for being the underlying mechanism of the word blindness suggestion, as they did not meet other criteria, such as reducing the RT of incongruent trials." This conclusion holds under the cold control theory, but not necessarily otherwise: suppose for instance that Highs after hypnotic induction do modify their first order processes, and not simply their metacognitive access to them (contra Cold-Control). Then, one could imagine that in this state they discover how to blur their vision in a much more efficient way than without hypnosis (in addition to doing that involuntarily). We would then have the same strategy (blurring), but performed at various levels of efficiencies.

p. 35 l. 39 I don't see the motivation for the reservation here expressed: "it still needs to be explored whether those from the full spectrum of hypnotisability can use this strategy to alleviate the interference, and whether the influence of imagination may generalise to other cognitive tasks, such as the flanker (Eriksen & Eriksen, 1974) or Simon task (Simon & Wolf, 1963)". The first part might be needed under the assumption of the cold control theory, but again, if one relaxes the constrain of cold control, it might simply be the case that Highs are able to achieve this feat of imagination, while Lows and Mediums would not. As for the last part, it seems not quite relevant to the precise discussion about the Stroop case.

p. 4 l. 18 "the by the"

p. 6 l. 34 McLeod

p. 16 ll 12- 15 trailing zeros could be removed.

p. 26 l. 30 "Table 2" -> 3?

p. 27 Figure 3 caption: the "Stroop interference scores" are computed with respect to both incongruent conditions, or only for the traditional semantic + response conflict condition?

I always sign my reviews

Jérôme Sackur

===PREPARING YOUR MANUSCRIPT===

===PREPARING YOUR REVISION IN SCHOLARONE===

- If you are providing image files for potential cover images, please upload these at this step, and inform the editorial office you have done so. You must hold the copyright to any image provided.
- A copy of your point-by-point response to referees and Editors. This will expedite the preparation of your proof.

- Ensure that your data access statement meets the requirements at <https://royalsociety.org/journals/authors/author-guidelines/#data>. You should ensure that you cite the dataset in your reference list. If you have deposited data etc in the Dryad repository, please include both the 'For publication' link and 'For review' link at this stage.
- If you are requesting an article processing charge waiver, you must select the relevant waiver option (if requesting a discretionary waiver, the form should have been uploaded at Step 3 'File upload' above).
- If you have uploaded ESM files, please ensure you follow the guidance at <https://royalsociety.org/journals/authors/author-guidelines/#supplementary-material> to include a suitable title and informative caption. An example of appropriate titling and captioning may be found at [https://figshare.com/articles/Table_S2_from_Is_there_a_trade-off_between_peak_performance_and_performance_breadth_across_temperatures_for_aerobic_sc ope_in_teleost_fishes_/3843624](https://figshare.com/articles/Table_S2_from_Is_there_a_trade-off_between_peak_performance_and_performance_breadth_across_temperatures_for_aerobic_scope_in_teleost_fishes_/3843624).

Author's Response to Decision Letter for (RSOS-202136.R0)

See Appendix A.

RSOS-202136.R1 (Revision)

Review form: Reviewer 4

Is the manuscript scientifically sound in its present form?

Yes

Are the interpretations and conclusions justified by the results?

Yes

Is the language acceptable?

Yes

Do you have any ethical concerns with this paper?

Yes

Have you any concerns about statistical analyses in this paper?

No

Recommendation?

Accept with minor revision (please list in comments)

Comments to the Author(s)

This aims to uncover the mechanisms of word blindness – a hypnotic phenomenon where high hypnotisable individuals can reduce the Stroop interference effect following a posthypnotic suggestion to experience word stimuli as gibberish. Here, the authors propose to test whether different explicit intentional strategies can replicate the pattern we observe during word blindness. This study presents interesting results that accounts for potential explanations of this phenomenon, and thereby tests a central assumption of cold control theory – namely that high hypnotisable individuals produce word blindness by voluntarily implementing one such strategy, while hypnosis yields poor metacognitive appraisal of their intentions to use this strategy.

This manuscript has already been reviewed. Here, I was asked to examine the revisions considering the comments made during the previous round of review. I only highlight the comments and revisions that I find the most relevant.

In my view, the authors addressed all the reviewers' comments to satisfaction.

Aside from the reviewers' comments, my only remark concerns one limitation to the overall rationale of the study with respect to CCT. Although this is a very minor point given the outcome, I nevertheless wonder if the current experiment provides a robust test for CCT. Given that Stroop interference suppression occurs mostly in highs and hardly in lows, if one strategy would have been a suitable candidate to explain word blindness, why would having poor metacognitive representations of intentions, per CCT, explain that highs are more likely to adopt this strategy compared to lows? Why would having good metacognitive representations of intentions prevent lows from behaving similarly to the highs?

Reviewer 1

Comment #1 and #2. Ok

Comment #3. I agree with the authors that the work from Augustinova and Ferrand (2012) undermines accounts of word blindness arguing for the suppression of semantic processing during the Stroop task. And while evidence tends towards the null hypothesis in both experiments regarding suppression of the semantic Stroop, none offer strong evidence, per the usual $<.33$ threshold adopted by most researchers in the field. Since they went to the trouble of estimating it, I recommend for authors to add the Bayes factors to the manuscript to provide all the information available to the reader.

Comment #4. Again, I concur with the authors, word blindness has already been replicated across several reports and different labs.

Comment #5. I agree with the authors. The relative differences between conditions we are investigating appears to make this comment somewhat irrelevant, unless there is more to the contingency effect that may interact with strategies – then it is not all else being equal.

Reviewer 2

Comment #6. I agree with the authors. This assessment is relevant insofar that variation of the Stroop interference effects across strategies and hypnotisability has important theoretical implications for CCT.

Comment #7 to #14. Ok

Comment #15. Again, I would recommend for this analysis to be included in the manuscript or supplementary material.

Comment # 16 to # 19. Ok

Comment #20. The decision to present the reduction of Stroop interference from baseline across strategies addressed the reviewer's about within-subject assessment.

Comment # 21. Ok

Reviewer #3

Comment #22. While the authors have made the underlying assumptions of CCT more explicit, the reviewer's comment brings to light a limitation to the rationale of the study. If one these strategies were to replicate the pattern in word blindness, the authors never address how CCT would account for the fact that low hypnotisable individuals fail to intentionally engage in the same strategies as high hypnotisable individuals? I fail to grasp how having accurate second-order representations of intention for lows (as opposed to inaccurate highs) account for the fact that word blindness only occurs for highs? This point comes to be irrelevant considering the results, yet remains pertinent for the rationale of the study.

Comment #23. Ok

Comment #24. Instead of adding the point about counterfactual in a footnote, I would include it in the main body of text since it hardly seems trivial and directly pertains to the main question of this research.

Comment #25 and #26. Ok

Comment #27. Again, this footnote might be easily missed.

Comment #28. I agree with the authors. The statistical tests already directly address the hypothesis. Additional tests about overall slowing down or speeding up are secondary with respect to the main hypothesis. Lastly, the omnibus test would seem redundant and uninformative at this point.

Comment #29 and #30. Ok

Comment #31. If they have not done so already, the authors should add this information to the methodology to help with future replications.

Comment #32 to #37. Ok

Comment #38. Similar comment as before: Adding this information to supplementary material may provide helpful information to researchers interested in using this method.

Comment #39. While this point is well taken with respect to CCT, knowing whether high hypnotisable individuals are better at implementing certain strategies could have been informative if these strategies offered a potential explanation to word blindness. The idea that at least some highs possess attentional abilities that could ease the implementation for some of these strategies speaks to a whole body of literature. Still, I largely agree with the authors that testing the relationship between the SWASH scores and self-reported ease of strategy implementation is

secondary with respect to the main goal of the study, and hardly seem relevant given that none of these strategies represent valid candidate for explaining word blindness and the absence of correlation between SWASH scores and reduction of interference effects across strategies.

Comment #40 to #44. Ok

Decision letter (RSOS-202136.R1)

Dear Dr Palfi,

On behalf of the Editors, we are pleased to inform you that your Manuscript RSOS-202136.R1 "Strategies that reduce Stroop interference" has been accepted for publication in Royal Society Open Science subject to minor revision in accordance with the referees' reports. Please find the referees' comments along with any feedback from the Editors below my signature.

Please submit your revised manuscript and required files (see below) no later than 7 days from today's (ie 09-Dec-2021) date. Note: the ScholarOne system will 'lock' if submission of the revision is attempted 7 or more days after the deadline. If you do not think you will be able to meet this deadline please contact the editorial office immediately.

on behalf of Dr Denes Szucs (Associate Editor) and Essi Viding (Subject Editor)
openscience@royalsociety.org

Reviewer comments to Author:

Reviewer: 4

Comments to the Author(s)

This aims to uncover the mechanisms of word blindness – a hypnotic phenomenon where high hypnotisable individuals can reduce the Stroop interference effect following a posthypnotic

suggestion to experience word stimuli as gibberish. Here, the authors propose to test whether different explicit intentional strategies can replicate the pattern we observe during word blindness. This study present interesting results that accounts for potential explanations of this phenomenon, and thereby test a central assumption of cold control theory – namely that high hypnotisable individuals produce word blindness by the voluntarily implementing one such strategy, while hypnosis yields poor metacognitive appraisal of their intentions to use this strategy.

This manuscript has already been reviewed. Here, I was asked to examine the revisions considering the comments made during the previous round of review. I only highlight the comments and revisions that I find the most relevant.

In my view, the authors addressed all the reviewers' comments to satisfaction.

Aside from the reviewers' comments, my only remark concerns one limitation to the overall rationale of the study with respect to CCT. Although this is a very minor point given the outcome, I nevertheless wonder if the current experiment provides a robust test for CCT. Given that Stroop interference suppression occurs mostly in highs and hardly in lows, if one strategy would have been a suitable candidate to explain word blindness, why would having poor metacognitive representations of intentions, per CCT, explains that highs are more likely adopt this strategy compares to low? Why having good metacognitive representations of intentions would prevent lows from behaving similarly to the highs?

Reviewer 1

Comment #1 and #2. Ok

Comment #3. I agree with the authors that the work from Augustinova and Ferrand (2012) undermines accounts of word blindness arguing for the suppression of semantic processing during the Stroop task. And while evidence tends towards the null hypothesis in both experiments regarding suppression of the semantic Stroop, none offer strong evidence, per the usual $<.33$ threshold adopted by most researchers in the field. Since they went to the trouble of estimating it, I recommend for authors to add the Bayes factors to the manuscript to provide all the information available to the reader.

Comment #4. Again, I concur with the authors, word blindness has already been replicated across several reports and different labs.

Comment #5. I agree with the authors. The relative differences between conditions we are investigating appears to make this comment somewhat irrelevant, unless there is more to the contingency effect that may interaction with strategies – then it is not all else being equal.

Reviewer 2

Comment #6. I agree with the authors. This assessment is relevant insofar that variation of the Stroop interference effects across strategies and hypnotisability has important theoretical implications for CCT.

Comment #7 to #14. Ok

Comment #15. Again, I would recommend for this analysis to be included in the manuscript or supplementary material.

Comment # 16 to # 19. Ok

Comment #20. The decision to present the reduction of Stroop interference from baseline across strategies addressed the reviewer's about within-subject assessment.

Comment # 21. Ok

Reviewer #3

Comment #22. While the authors have made the underlying assumptions of CCT more explicit, the reviewer's comment brings to light a limitation to the rationale of the study. If one these strategies were to replicate the pattern in word blindness, the authors never address how CCT would account for the fact that low hypnotisable individuals fail to intentionally engage in the same strategies as high hypnotisable individuals? I fail to grasp how having accurate second-order representations of intention for lows (as opposed to inaccurate highs) account for the fact that word blindness only occurs for highs? This point comes to be irrelevant considering the results, yet remains pertinent for the rationale of the study.

Comment #23. Ok

Comment #24. Instead of adding the point about counterfactual in a footnote, I would include it in the main body of text since it hardly seems trivial and directly pertains to the main question of this research.

Comment #25 and #26. Ok

Comment #27. Again, this footnote might be easily missed.

Comment #28. I agree with the authors. The statistical tests already directly address the hypothesis. Additional tests about overall slowing down or speeding up are secondary with respect to the main hypothesis. Lastly, the omnibus test would seem redundant and uninformative at this point.

Comment #29 and #30. Ok

Comment #31. If they have not done so already, the authors should add this information to the methodology to help with future replications.

Comment #32 to #37. Ok

Comment #38. Similar comment as before: Adding this information to supplementary material may provide helpful information to researchers interested in using this method.

Comment #39. While this point is well taken with respect to CCT, knowing whether high hypnotisable individuals are better at implementing certain strategies could have been informative if these strategies offered a potential explanation to word blindness. The idea that at least some highs possess attentional abilities that could ease the implementation for some of these strategies speaks to a whole body of literature. Still, I largely agree with the authors that testing the relationship between the SWASH scores and self-reported ease of strategy implementation is secondary with respect to the main goal of the study, and hardly seem relevant given that none of these strategies represent valid candidate for explaining word blindness and the absence of correlation between SWASH scores and reduction of interference effects across strategies.

Comment #40 to #44. Ok

===PREPARING YOUR REVISION IN SCHOLARONE===

-- If you are requesting an article processing charge waiver, you must select the relevant waiver option (if requesting a discretionary waiver, the form should have been uploaded, see 'File upload' above).

-- If you have uploaded any electronic supplementary (ESM) files, please ensure you follow the guidance at <https://royalsociety.org/journals/authors/author-guidelines/#supplementary-material> to include a suitable title and informative caption. An example of appropriate titling and captioning may be found at https://figshare.com/articles/Table_S2_from_Is_there_a_trade-off_between_peak_performance_and_performance_breadth_across_temperatures_for_aerobic_scope_in_teleost_fishes_/3843624.

Author's Response to Decision Letter for (RSOS-202136.R1)

See Appendix B.

Decision letter (RSOS-202136.R2)

Dear Dr Palfi,

I am pleased to inform you that your manuscript entitled "Strategies that reduce Stroop interference" is now accepted for publication in Royal Society Open Science.

Please see the Royal Society Publishing guidance on how you may share your accepted author manuscript at <https://royalsociety.org/journals/ethics-policies/media-embargo/>. After publication, some additional ways to effectively promote your article can also be found here

<https://royalsociety.org/blog/2020/07/promoting-your-latest-paper-and-tracking-your-results/>.

on behalf of Dr Denes Szucs (Associate Editor) and Essi Viding (Subject Editor)
openscience@royalsociety.org

Appendix A

Dear Dr Denes Szucs and Essi Viding,

My co-authors and I are happy to submit a revised version of our manuscript "Strategies that reduce Stroop interference" (ID RSOS-202136) for publication in Royal Society Open Science.

We would like to thank you, and the reviewers for their constructive suggestions for improvement and for the chance to revise and resubmit our project. Below, in bold, is a point-by-point response to how we have changed the manuscript in order to accommodate the recommendations. We also highlighted all changes in our Manuscript and Supplementary Materials.

We look forward to your comments on this revision.

Kind regards,
Bence Palfi, on behalf of the authors

Reviewer comments to Author:

Reviewer: 1

Specific points

Comment #1

• Typos. There are a couple of places where a word was struck out and the strike-out mark remained: p.21, "how" is struck out and replaced by "whether": p.31, "good" is struck out and replaced by "strong"

Reply #1

Corrected

Comment#2

• Typos. P.26, end of first para, where it says "Table 2", I'm reasonably sure the authors mean Table 4.

Reply#2

Corrected

Comment #3

• p.20 bottom, "Another key characteristic of the word blindness suggestion is that it seems to reduce interference by attenuating response competition and not by de-automatising reading per se". This claim is just stated, and a bunch of references listed "(Augustinova & Ferrand, 2012; Palfi Parris, Seth, & Dienes, 2018; Parris, Dienes & Hodgson, 2013; contrast Raz, Fan & Posner, 2005)", without describing the evidence presented by these studies. I am familiar with the Augustinova & Ferrand (2012) study in which the hypnotic suggestion of wordblindness reduced the standard Stroop interference effect with response-incongruent words but not the interference produced by color-associated words (e.g., "LEMON" presented in blue). But as the present authors point out later in Footnote 3 (p.22), the interference effect produced by the color-associated words is small and not robust (Kinoshita, Mills & Norris, 2018), so this result may just indicate a lack of sensitivity to the manipulation, rather than evidence that the wordblindness suggestion reduces response competition, but not semantic conflict. I am not familiar with the other studies by the authors' group (Palfi, et al., 2018 – the References indicates this is a ms in preparation; Parris et al.,

2013), and in any case the evidence needs to be described here to provide the rationale for testing this claim in Experiment 2. This is particularly important as it appears that the claim that the hypnotic suggestion works by reducing response competition does not have universal support (“contrast Raz, Fan & Posner, 2005”).

Reply #3

The core claim that the word blindness suggestion leaves the semantic conflict component of Stroop interference intact is in fact universally accepted. It is not contested by the research group of Raz (e.g., Lifshitz et al., 2013, *Cortex*; Landry, Lifshitz, & Raz, 2017, *Neuroscience and Biobehavioural Reviews*) and they acknowledge that this finding is problematic for their pure de-automatisation account. Consequently, they modified their model of the word blindness suggestion and do not expect it to “fully suppress reading and semantic processing” (Landry, Lifshitz, & Raz, 2017, *Neuroscience and Biobehavioural Reviews*). Importantly, the cited Raz et al. study (2005) does not provide a direct test of the de-automatisation account. It reports neural correlates of the word blindness suggestion from a small (8 highly suggestible people) study. ERP data suggest that early visual input may be dampened by the suggestion (which is in line with de-automatisation) but it also suggests that suggestion may reduce cognitive conflict at a later stage (which is in line with both accounts). Moreover, the behavioural data show that the suggestion reduced RTs of all trials, including the congruent trials, which conflicts with the idea that the participants did not read the words due to the suggestion (see also Comment #10 by Reviewer 2 for the same argument). The bottom line is that the available evidence is more in line with the response competition account and this (to some extent) is acknowledged by the researchers who put forward the alternative account. We’ve revised the intro of the second experiment to improve the clarity of the argument.

Regarding the sensitivity argument concerning the semantic conflict produced by colour-associated words: whilst it is true that semantic associative Stroop conflict is not robust in the manual Stroop task reported in Kinoshita et al., it has been robust in other studies (e.g. Augustinova, Parris, & Ferrand, 2019). Moreover, as we argued in the manuscript, Augustinova and Ferrand also used a vocal Stroop task. More importantly, the lack of sensitivity is an empirical claim that can be tested using the data provided by A and F. We calculated Bayes factors for the model predicting that the word blindness suggestion should reduce semantic conflict (a model that is in line with the de-automatisation account but not with the response competition account). We found a $BF_{H(0,10)}$ of 0.42 in Experiment 1 and a $BF_{H(0,10)}$ of 0.39 in Experiment 2. These imply that we have Bayesian evidence in support of the model predicting no effect on semantic conflict (i.e., the response competition model is supported).

Concerning the other behavioural studies cited by us that back the response competition account of word blindness suggestion. We have added a brief description of Parris et al. 2013’s argument for the response competition account. However, these studies are not directly relevant to the application of the non-response set trials, since they applied different approaches to compare the de-automatization and the response competition models. Therefore, we only refer to them as additional behavioural evidence for the response competition account, but do not expand on them to avoid having an unnecessarily long Intro to the second experiment. The core

argument here is that semantic conflict should not be reduced (and certainly not eliminated) by a strategy that is used to execute the word blindness suggestion, and we believe that the A and F paper provides evidence to back this claim.

Comment #4

• I note that some of the Stroop results reported in the literature as being immune to the hypnotic suggestion of wordblindness are not factually correct. For example, Raz & Campbell (2011) reported that while the hypnotic suggestion reduced the overall Stroop interference effect it did not affect the negative priming effect. We (Mills, Kinoshita & Norris, 2019, *Frontiers in Psychology*, “No negative priming effect in the manual Stroop task”) argued that this claim likely reflects the incorrect way in which the negative priming effect is calculated in the study (confounded with the response repetition effect), as the manual Stroop task generally does not produce the negative priming effect in the first place. So I think it is really important to replicate the pattern of Stroop results supposedly produced by the hypnotic suggestion of wordblindness reported in the literature before investigating what mechanism explains the pattern of results.

Reply #4

The word blindness effect (reduction in Stroop interference by the suggestion) is a well-established phenomenon that have been replicated several times by at least four independent research labs (p. 4). We believe that investigating the potential mechanism of an established phenomenon is a valid scientific quest, and that the success of a study should be evaluated on its ability to test these mechanisms rather than on its results (i.e., finding evidence against four plausible strategies does not indicate that the study is unsuccessful).

In the present study, the pattern of results we expected to see (reduction in incongruent RTs, no reduction of semantic conflict) by the strategies that could explain the word blindness effect were the ones that we found sufficiently grounded (see also Reply #3). As far as we are concerned, the relationship between the word blindness effect and negative priming was only investigated once (in the cited study) and even then, it was done as an admittedly post-hoc and underpowered analysis (moreover, as the reviewer argues, it was done incorrectly). Therefore, negative priming has no special relevance to our study.

Comment #5

• Contingency. Recent Stroop studies have pointed out that the congruent condition (e.g., the word RED presented in red colour) and the incongruent condition (e.g., the word RED presented in green colour) differ in the word-colour contingency when there are more than 2 response colours and the number of congruent trials and incongruent trials are equated, as was done here, unless only one incongruent colour was used with an incongruent word. It is not clear from the description if this is what was done here in Experiment 1. Please make this clear. Contingency has a large effect on Stroop interference (response to color is much faster for high-contingency items (e.g., Lin & MacLeod, 2017, *JEP:LMC*).

Reply #5

We have clarified in the manuscript that the incongruent words could be presented in any of the mismatching colours (p. 10). This matches the arrangement reported in the original Raz et al. (2002) study and all subsequent WBSE studies. In none of these studies was contingency controlled. Nevertheless, contingency learning, whilst an

important issue, imposes little threat to the conclusions of the current paper since contingency was identical in the No strategy and Strategy conditions. In fact, some researchers include neutral trials in the Stroop task to mitigate or circumvent the issue of contingency learning, so that they can test the presence of a pure congruency effect (e.g., Lorentz et al., 2016, *Frontiers in Psychology*).

Reviewer: 2

Comment #6

The introduction establishes two goals for the manuscript: (1) by what “strategy” does the word-blindness effect arise and (2) is the strategy related to hypnotisability. The first of these goals is clear and easy to follow; the latter not so much. Perhaps the authors could re-organize the introduction so that all of the discussion with regard to strategies is dealt with first and then delve into the second goal? That said, I’m not sure the experiment can even address the theoretical motivation for latter goal. For instance, is there an outcome that would actually falsify the account?

Reply #6

The second question of the paper is less important to the understanding of the word blindness suggestion but it still holds theoretical relevance to cold control theory (which is in the focus of the paper) and practical relevance to the test of the strategies. The practical relevance is that under this assumption (i.e., highs and lows differ in metacognition but not in cognitive control) we could recruit people from the whole range of hypnotisability and so it is easier to have a well powered test of the strategies. This assumption is already supported by empirical evidence (e.g., Dienes, et al., 2009; Parris, 2017; Varga, Németh, & Szekely, 2011), but it can also be simply tested by the proposed correlation analysis. If we were to find a positive correlation between hypnotisability and strategy effectiveness, then the assumption would be disconfirmed, and we would need to test the strategies only on highs. We have made all of this clear in the last paragraph of the Introduction and we made sure that we do not raise these issues in earlier paragraphs. We also explain in the Introduction of the second experiment why we do not need to check this again, and why we continued to recruit people from the whole range of hypnotisability.

Comment #7

According to the authors, there are two (maybe three) signature effects that need to be observed by a strategy if it is going to provide a reasonable explanation of behaviour under word blindness conditions with highly suggestible participants. The first signature is a reduction in Stroop Interference. The second signature is a speed up on interference trials (as opposed to a slowdown across neutral and congruent trials). A possible third signature, which the authors admit is more variable is the absence of a reduction in Stroop facilitation. The situation surrounding the latter should probably be unpacked a little more (an additional sentence would be enough and I’m not suggesting that the authors need to test this). My concern is that the hypotheses/ predictions of Experiment 1 do not reflect the earlier purported importance of these two effects. In addition, the second signature is not described as a critical test in the results of Experiment 1. This stands in contrast to the analyses of Experiment 2, where they are both clearly emphasized. And again the

conjunction of these two effects are focus of the General Discussion.

Reply #7

We have added the second signature effect to the sentence in which we describe our hypotheses/expectations for Exp1 (p.9). Also, to make the analyses of Exp1 and Exp2 consistent, we highlighted the test of the reduction of incongruent RTs as a crucial test in Exp 1, which is separate from the crucial test of the reduction of Stroop interference.

We did not define a signature test involving Stroop facilitation as the current empirical evidence is ambiguous. We have clarified this in a footnote (p.9).

Comment #8

The flow of the manuscript with regard to the other goal of the paper (testing a (possible???) prediction of cold-control theory. The authors start with this issue, leave to discuss strategies, then return to the theory of cold-control theory towards the end of introduction (the motivation for goal 2, which is abandoned after Experiment 1 and really doesn't play any necessary role in the manuscript).

Reply #8

The discussion of cold control theory at beginning of the paper is to address the core justification of the study. We test intentional strategies as simple theories of hypnosis (such as cold control theory) predict that this is how highs reduce Stroop interference. If we cannot identify such a strategy, then more complex explanations of hypnosis may be needed to address the word blindness suggestion. We clarify this in the Introduction and the Discussion.

Regarding the secondary goal of the manuscript, see our Reply #6.

Comment #9

In the general discussion, I'm not sure where the statement that "these strategies seemed to attenuate Stroop interference by affecting the general speed of responses" comes from. The authors quickly move on to clarify what they mean, but if you are going to do this, why include a description of a main effect that is never tested for and clearly doesn't explain the data.

Reply # 9

We highlighted this feature of the data as it may help us understand how looking-away and blurring managed to reduce Stroop interference (as we argue in the discussion, the general slowing down of responses may be how these strategies equalise responses to different congruency trials). The reviewer correctly points out that we did not include a formal statistical test of the main effects, and we only referred to the Tables presenting average condition RTs. We have added the test of the main effect of general speed (incongruent + neutral RTs) to clarify that there is evidence suggesting that the successful strategies slowed down rather than speeded up the RTs.

Comment #10

I recommend that the authors discuss a few more findings from the Stroop literature. In part because I think it would be worthwhile for the authors to articulate that the word blindness effect could be mislabelled, if it is thought of as an explanation as opposed to the instructions

given to participants. In my mind, one issue is whether the strategies examined in the manuscript can explain the word blindness effect (they do not), whereas another issue is whether the word blindness induction is affecting reading at all. For instance, reducing readability should affect both facilitation and interference, moving both towards the neutral condition – especially if the neutral condition is not a word. With this in mind, a paper by Melara and Mounts (1993) on dimensional discriminability may be relevant. Further, it may be worthwhile discussing factors that affect both facilitation and interference, such as the response set size (I think MacLeod, 1991 might provide a discussion of this), and factors that affect only interference such as the proportion of congruent/incongruent trials, which Lowe and Mitterer (1982) suggested specifically targets response conflict resolution.

Reply #10

The current manuscript does not aim to provide arguments for or against the de-automatisation of reading and the response competition models of the suggestion; we aimed to test whether any of the strategies previously identified in the literature as leading to a reduction in Stroop interference can produce the patterns in the RT data associated with the Word Blindness Suggestion Effect.

The term *word blindness* is used by us to refer to the instructions of the suggestion and the experience reported by highs when they respond to the suggestion (i.e., blindness to the meaning of the words) and does not mean to imply the mechanism by which it operates (i.e., turning off reading); hence the name being the ‘word blindness *suggestion* effect’ and not the ‘word blindness effect’. In fact, we discussed these issues in another manuscript (Parris et al., 2013) and in a more recent paper of ours, in which we apply a proportion incongruency manipulation to differentiate between the two models of the suggestion (Palfi, B., Parris, B. A., Seth, A. K., & Dienes, Z. (2018). Does unconscious control depend on conflict?. (Manuscript in preparation. Preprint: <https://psyarxiv.com/a68js/>). Nonetheless, we thank the reviewer for recommending the cited papers and we agree that these are relevant to the understanding of the suggestion, however, we believe that these papers better fit the other manuscript of ours.

Comment #11

The authors might also consider discussing a paper by Reynolds, Kwan & Smilek (2010) when discussing factors that modulate the Stroop effect at the very beginning of the manuscript. This is an oft neglected paper that goes some way towards examining the bottom up processes that modulate the Stroop effect and might provide a useful framework for thinking about single letter colouring and how the word blindness could affect processing. However, I think this might be better served in the introduction.

Reply #11

We have included this paper in the list of the citations about the bottom-up processes that influence the size of the Stroop effect.

Comment #12

One other element from the Stroop literature that should be elaborated on is the issue of a “neutral” condition. In particular, I feel that the discussion of neutral RTs in the General Discussion cannot occur without a discussion of the problems with the concept of a neutral condition. In particular the choice to use neutral words instead of over possible baselines.

Reply #12

We agree with the Reviewer that the choice of the neutral words is an interesting topic

as it influences the extent of the Stroop interference effect. However, our experimental design used the same neutral trials as the original Raz study and many follow-up studies testing the word blindness effect rendering this issue irrelevant for the central question of the current study. Moreover, a recent study (Zahedi et al., 2019) included multiple neutral conditions (non-word letter strings, high- and low-frequency words) and found that the suggestion affected Stroop interference for each of them.

Comment #13

In the methods: What is the size of the screen? The look away location could make it difficult to perceive the color. For instance, a 20 inch screen would mean that they were looking approximately 25 degrees from the colour. I realize that participants were told that they didn't have to look in the corner if it was too far, but still this info is relevant.

Reply #13

We added info about the computer screen sizes to the Methods sections. In the first Experiment we used a 15.6-inch screen, whereas in the second experiment we used an 18-inch screen.

Comment #14

I understand the value of replicating the methods reported by Raz et al. (2002) however, some deviations would have been nice. For instance, if the hypnotic suggestion effect is largest at short RSIs why not include or use that condition? See also p31 of the discussion.

Reply #14

The RSI of the present study was substantially shorter (approximately with 1 second) compared to the RSI of the Raz et al. (2002) study, which had a fixed ISI of 4 seconds.

Regarding the manipulation of RSI. This effect was only tested by a single study (Parris et al., 2012) whose key result was put into doubt via a later distributional analysis (Parris et al., 2013) so it is a less accepted signature effect of the word blindness suggestion than the ones we tested. We saw no value in adding an additional between subjects' condition to Exp 2 (on top of the test of the semantic conflict) when the strategies already failed to pass stronger/more accepted signature tests.

Comment #15

More troubling was their decision to not label the response keys (Raz et al., 2002 did label them). The lack of substantial practice (36 trials vs. 32 in Raz et al.) means that the participants were probably still learning the key-mappings well into the first block. This has implications for their analysis of the block 1 data.

Reply #15

Colour labelling would invite the use of a potential colour-matching strategy, which we have controlled for here by not using these labels. Given the purpose of the current study, controlling for this potential strategy was more important than providing a visual aid for the participants. We have added a comment about this in the manuscript. Also, participants of the present study had a longer (admittedly with only a few trials) practice session than the participants of the Raz et al. study.

Moreover, if the participants had not had sufficient practice and they were still learning response mapping during the first block, then we would expect that they

would respond substantially slower (and they would commit more errors) in general during the first block than during the rest of the blocks (note that there is an emphasis on substantial here as a small difference between the first block and the following blocks is expected even if response mappings are well practiced). We did this analysis focusing on the no strategy condition. We ran multilevel linear regressions with congruency and block order (either 1 or 2, where 2 means that the block was not presented at first) as predictors, random intercept by subjects and RT (and error rates) as the dependent variable. The main effect of block order remained non-significant for both experiments, and the Bayesian analyses were insensitive (the only analysis that yielded evidence leaning towards H1 was the analysis of RTs in the first experiment).

Exp1:

RT: $b = 47 \text{ ms}$, $t(74.30) = 1.50$, $p = .139$, $B_{H(0, 30)} = 2.09$

ER: $b = -1.24\%$, $t(129.12) = -0.846$, $p = .399$, $B_{H(0, 1.5)} = 0.46$

Exp2:

RT: $b = 10 \text{ ms}$, $t(44.40) = 0.254$, $p = .801$, $B_{H(0, 30)} = 0.90$

ER: $b = 0.72\%$, $t(65.64) = 0.39$, $p = .700$, $B_{H(0, 1.5)} = 0.95$

Comment #16

The analyses of “carry-over” effects, in which the focused solely on the data from Block 1, is not really an analysis of carry-over effects. Instead, it is an analysis without carry-over effects. I realize that this is a minor detail, but it is worth pointing out. Further, this given the lack of practice the interpretability of the first block problematic. For instance, if participants are still learning the response mapping, this could interfere differently with each of strategy manipulations. Further, the authors do not report what the sample size is per condition – strategy was randomly assigned to block. Finally, this analysis is a between subjects comparison. Therefore, the power to detect any effect would be weak at the best of times, let alone with all of these problems. Although there is no formal analysis of this data in the manuscript the authors do draw conclusions from the data, which are presented in a table. I recommend that the authors remove the analysis or tone down their conclusions to adequately reflect the interpretability problems.

Reply #16

As we mentioned in the reply above, the evidence for the lack of practice is insensitive. That being said, we agree with the reviewer that the provided data represent a without carry over effect scenario rather than the effect of a potential carry over itself. As we don't find this analysis critical to the main arguments, we moved it to the Supplementary materials and we do not draw conclusions from the data anymore (note that we also added the number of subjects per condition).

Comment #17

Analyses

I should note that I did not focus on the Bayesian analyses in too much detail. That said, I did note that it does not appear that the authors dealt with how observing an effect in the opposite direction affects Bayesian calculations.

Reply #17

Most of our models representing the predictions of the alternative hypotheses were directional as these alternative hypotheses had clear directional predictions. We did not run analyses testing models with predictions towards the opposite direction as we could not link these models to meaningful theories (e.g., no relevant theory expects an increase in Stroop interference by the strategies). That being said, the fact that the looking-away and blurring strategies increased rather than decreased incongruent RTs may be interesting for future research aiming to unravel how these strategies work. Therefore, we have now reported Bayes factors that tested H1s that predicted the increase (rather than decrease) of incongruent RTs by the strategies. In experiment 1, evidence remained insensitive, whereas in experiment 2 we found moderate evidence that both strategies increased incongruent RTs compared to the no strategy condition.

Comment #18

There are also some missing effect size values in the analyses reported on page 25 of the manuscript.

Reply #18

Corrected.

Comment #19

I am blown away by the size of the Stroop effects in the present study. They are absolutely huge given the methodology.

Reply #19

We reported a table in the Supplementary materials (S5) containing the condition means in RTs of all published studies between 2002 and 2019 that tested the word blindness effect. The reported mean Stroop effect in the no suggestion condition was 124ms and the Stroop interference effect was 89ms. In contrast, the size of the Stroop effect in the no strategy condition in Exp 1 was 126ms (130ms in Exp 2) and the size of the Stroop interference effect in Exp 1 was 78ms (79ms in Exp 2).

Comment #20

I am not a huge fan of violin plots although I recognize that they are all the rage lately. My concern is that they are largely useless in within-subjects designs as the correlation between conditions is critical to their interpretation. This is also true of the standard deviations in the tables and confidence intervals in graphs – see Masson & Loftus.

Reply #20

The reviewer is correct in that the provided violin plots are not ideal to complement the within-subjects statistical tests. Therefore, we modified them to be in line with the correlational plot (Figure 2), which used reduction in Stroop interference (difference between no strategy and a strategy condition) rather than Stroop interference per condition. This way, the violin plots represent the same aspect of the data than the corresponding statistical tests (Crucial test 1 in both experiments). Regarding the Tables, we believe that it is felicitous to keep them so that condition means and SDs can be compared to those of the earlier publications in the field (most publications investigating the word blindness effect reported the condition means and SDs). Also, we reported the raw effect sizes and their 95% CIs regarding the contrasts of the conditions in the main text so the readers can easily find them.

Comments #21

Minor comments

p4, ln 35. The word to should be removed.

p4, ln 50. I'm not sure if it is worth mentioning, but the stroop effect is robust at a group level but not at the level of individuals.

p6, ln 8. the Bowers (1990) reference is missing

Reply #21

Corrected. We clarified that by robustness we mean that the Stroop effect emerges for every individual rather than the size of the Stroop effect is robust at the level of individuals.

Reviewer: 3

Comment #22

Comments to the Author(s)

A "word blindness" suggestion can decrease the Stroop interference effect, when given to highly hypnotisable participants. The paper argues that, within the context of the cold control theory (CCT) of hypnosis (according to which hypnosis removes metacognitive awareness of the strategy at work), four plausible strategies could be employed. It reports that two of them "looking away" and "blurring" are efficient, but that they do not exactly reproduce the pattern of results obtained with the word blindness suggestion. Hence the author conclude that the strategy employed by participants under a word blindness suggestion has yet to be found.

The paper is clear and provides a useful contribution to the current debates about the mechanisms of hypnosis. I have a few general remarks:

1/ At times, the authors might seem to forget that theirs is a test of the conjunction of the CCT and of some particular strategies. It might be for instance that the CCT is not the complete story, and that under hypnosis Highs do things a bit differently than outside hypnosis. For instance, It might be that under hypnosis, Highs "look away" in a specific manner, which they are not able to reproduce when given the explicit instruction to look away without hypnosis. Thus the inadequacy of the strategies is most meaningful under the assumption of the CCT (and there are many reasons to accept this theory, but the present paper is not one of them, so the assumptive nature of the CCT should not be hidden or forgotten).

Reply #22

We made sure that the assumptive nature of the CCT is explicit in the manuscript by adding a paragraph in the discussion about this issue. We also highlighted this at more points in the manuscript.

Comment #23

2/ I think the discussion about what it would mean to take the word blindness suggestion literally should not be deferred to the Discussion. Otherwise, it seems that the authors take it for granted that the suggestion is necessarily metaphorical. On p. 4, the authors seem to take it as self-evident that a word blindness suggestion of the form "you will see the words as gibberish" is not directly actionable by participants. (As if you would say: "levitate and touch the ceiling", and that some participants implement that with the "grab a chair" strategy, some with the "jump and extend your arms" strategy, etc...). One might argue that it is intrinsic to hypnotic suggestions to be obscure, ambiguous, metaphorical, etc. But it is not necessarily so, and if this is the authors' stance, they should argue for it. I think that at least, the authors need to present some argument in the Introduction as to why they don't test the literal content of the suggestion as a strategy.

Reply #23

Our reason is more practical. We addressed this possibility in another paper of ours as we argued in the general discussion (Palfi et al., 2020, Cortex). We have made this clear in the introduction in a footnote and we refer the reader to the general discussion and the other paper.

Comment #24

Furthermore, when the authors do address this issue in the Discussion, they interpret it as the imagination of a counterfactual world (ie, participants would imagine a world in which they have dyslexia). But there might be an even more literal interpretation, along the lines of the concepts of "aspect switching" and "seeing as" as they are discussed in aesthetics and philosophy (cf notably Wittgenstein, /Philosophical Investigations/, /Remarks on the philosophy of psychology/). To see a particular cloud as the head of a cat, one does not need to imagine a counterfactual world in which bodyless giant cats fly.

Reply #24

The "seeing as" scenario described by the Reviewer does not align well with what highs generally say they experience when they are given hypnotic suggestions. Importantly, seeing a particular cloud as similar to a cat is not the same as genuinely seeing it as a cat. The reviewer is correct that the former can be done without imagining bodyless giant cats flying around but the latter would require us to hold a counterfactual world. In the case of the word blindness suggestion, the question is whether highs see something similar to a foreign word, they knowingly see an English word that is similar to a foreign one - or do they see a foreign one. The latter can be achieved by imagining a counterfactual world in which it is a foreign word, then not being aware one intended to construct the counterfactual world (e.g., according to cold control theory). The latter would produce the stronger phenomenology and, crucially, it is closer to the experience reported by highs (e.g., Palfi et al., 2020, Cortex). We now explain this and related issues in the discussion.

Comment #25

3/ The systematicity of the statistical analysis is of course a good thing, but it sometimes makes the results hard to follow. Below I detail some of my difficulties.

* Introduction

p. 6 l. 52 When referring to attention, here I would prefix it with "spatial". Because, obviously, if any strategy is to be efficacious, it is a form of attentional strategy, although not necessarily spatial.

Reply #25

We thank the reviewer for the suggestion and added the word *spatial* to the text.

Comment #26

p. 9 I'm not sure the last short paragraph of the Introduction is really necessary.

Reply #26

We agree with the reviewer, and we removed the last paragraph.

Comment #27

* Stimulus and apparatus

No detail is provided concerning the psychophysical properties of the colors. Values in the RGB space are not very informative, but it would be nice to have at least subjective judgments. In my experience, with "out of the box" software colored stimuli, yellows make for really less legible words. (Acknowledgely, it is orthogonal to the hypotheses, but still...) This is perhaps more important in Exp. 2, where two color sets are employed.

Reply #27

When piloting the study, we had the same experience as described by the reviewer and found that the default yellow colour presented against a white background is not as readable as the other colours. Hence, we did not use the recognised colour keyword `yellow` and replaced it with a slightly darker colour (hex colour code #ffef36) to make the words presented in yellow more readable. We added the hex colour codes of the used colours to two footnotes in the manuscript.

Comment #28

p. 12 Statistical Analyses

I understand the logic of analyzing difference scores, but I think raw RTs would still be informative, and indeed you report them in table 1. You do have rather large between condition differences in mean speed. I think they are interesting as such. Indeed one of the leitmotiv of the analyses is whether such and such strategy "reduces the RTs of incongruent trials" (cf p. 16 l34 for instance). Accordingly, sometimes the difference scores are computed across congruent and incongruent trials within instruction condition, and compared with the

analogous difference within the no instruction condition; sometimes they are computed between conditions within the incongruent trials. Essentially, if I understand well, this could all be summed up in an ANOVA model with two main effects (condition and congruency) and an interaction of condition and congruency. To me such a presentation of the results would be much clearer, followed by the Bayesian t-tests for the contrasts of interest.

Reply #28

The reviewer is correct that running 2x2 ANOVAs (no strategy vs one of the strategies, incongruent vs neutral trial) for all strategies would also produce the wanted tests (note that we are always interested in the neutral rather than congruent trial, so the congruent trials are not included in any of the analyses apart from the test of the traditional Stroop effect). In fact, for the test of the interaction between strategy and congruency, this is what we have done as the Bayesian t-test on the difference scores (Crucial test 1) is identical to a 2x2 F-test (e.g., Abelson, 1995, Statistics as principled argument).

We agree with the reviewer that the main effect of condition (strategy) on the general speed of responses can be interesting, so we added this analysis as a supporting test of interest. Again, we conducted this by comparing each strategy condition to the no strategy condition, and only considering incongruent and neutral responses.

If the Reviewer meant that we should run a 5x3 ANOVA including all strategy conditions and congruency conditions, then we argue that the ANOVA approach would not be able to answer the questions asked by the current study. The hypotheses put forward by the study can only be tested by direct comparisons of strategy and no strategy conditions, so we see no merit in running an omnibus ANOVA.

Comment #29

p. 13 l23: I don't understand the reference of the "60ms and 105ms, respectively". From the next sentence, I infer that 60ms is the prior for the Stroop interference, but what is 105ms?

Reply #29

The Reviewer is correct, 60ms is the prior of the Stroop interference effect (incongruent – neutral RT). 105ms is the prior of the traditional Stroop congruency effect (incongruent – congruent RT). We have now clarified this in the text.

Comment #30

* Design and procedure

The time sequence of the trial is a not really precise. Am I right to infer that the stimulus has a fixed duration of 2 s. yet the inter-stimulus interval is variable?

Reply #30

The stimulus was presented for 2 s or until a response button was pressed. We've corrected this in the methods. The ISI was indeed variable; however, the RSI was fixed.

Comment #31

p. 12 / Appendix: The strategies must have been explained at first before the experiment started? Otherwise, what were participants to make of the "no-strategy" instruction "This time do not use any of the strategies we have instructed you in previous blocks." in case they happened to start with a "no-strategy" block?

Reply #31

They were told at the beginning of the experiment that they will be given strategies in some conditions, but the strategies were not specified.

Comment #32

p. 12 / Appendix Why is it that the "No-strategy" instruction is the only one to have a speed and accuracy instruction ("as quickly and as accurately as you can")? I don't expect that to have a huge impact, but still, one could argue that this creates a self-evaluation pressure (and indeed, the no-strategy condition is the fastest of all.)

Reply #32

Speed and accuracy were equally emphasised at the beginning of the study, which we've made clear in the methods section now. We kept this message in the no-strategy condition so that we could provide them the same instructions that we gave to participants in the no suggestion condition of studies including a word blindness suggestion condition. It is unlikely that the participants were not trying their best in other conditions as they were the quickest in the Goal maintenance strategy condition (see Table 1) and error rates were comparable across conditions.

Comment #33

* Results

I feel that an error rate of 8.2% is not totally negligible, on account of the slow pace of the experiment. I think it might be valuable to test whether there are congruency effects in the error rates, and whether they are modulated by the strategies.

Reply #33

Now, we report these analyses in the Supplementary materials. In short, we found evidence for the Stroop interference effect (incongruent vs neutral trials), but the evidence regarding the influence of the strategies remained insensitive, mostly leaning towards no effect.

Comment #34

p. 14 l44 "The error rates were low..." it must be the proportion of outliers, not the errors?

Reply #34

Corrected.

Comment #35

p. 15 "Crucial test 1" This paragraph demands that the reader maintain the information that the "Stroop interference effect" (resp.

the "Stroop effect") is the difference between incongruent and neutral trials (resp. incongruent and congruent).

Reply #35

We added after Stroop interference that it is “(incongruent RTs – neutral RTs)” to ensure that the readers follow that all crucial analysis ignore the congruent trials.

Comment #36

p. 19 The legend of Table 2 is not specific enough. I think it would be important to know how many participants are included in each row of the table. If I understand well, that number would be around 12 (57/5)?

Reply #36

We added a column indicating the number of participants in each row. We did the same for Table 4. Moreover, we moved the carry-over effect section including these tables to the Supplementary materials (see reply #16).

Comment #37

* Experiment 2

** Results

Same remarks as above about the error rate and outlier proportions.

Reply #37

Corrected

Comment #38

p. 25 With respect to the comparison of the response sets A and B, I would guess that the colors of the original one are much more saturated and luminous, which might explain why there is a small difference between the two sets? I surmise that equiluminant colors would help in reducing the difference.

Reply #38

In fact, there was virtually no difference between the two sets regarding the Stroop interference effect (incongruent RTs – neutral RTs): Set A = 78ms, Set B = 79ms. Set A and B numerically differed in terms of the size of the semantic Stroop effect (non-response set incongruent RTs – neutral RTs), however, this difference was the other way around than one would expect based on the explanation offered by the reviewer. Colours were slightly more saturated and luminous in Set A so one would expect that the semantic Stroop effect is smaller in that set than in Set B. Yet, we observed that it was 49ms in Set A and 15 ms in Set B (Note that the Bayes factor was 1.08 so the evidence regarding the difference was completely insensitive).

Comment #39

* General Discussion

It would be interesting to discuss the ease of each strategy (as reported subjectively by participants) with respect to the cold

control theory of hypnosis. It may be that this value is correlated with SWASH score...

Reply #39

The proposed analysis would be interesting if we were to introduce a hypnosis condition as cold control theory (and many other theories of hypnosis) expects that highs experience their responses to suggestions as involuntary, something that happens by itself. However, this question concerns the alteration of subjective experiences by hypnosis, which is irrelevant to the central theme of the current paper: objective reduction in Stroop interference by voluntary strategies, which may explain the word blindness effect.

Comment #40

The authors do not envision the possibility that each individual participant may have a preferred (most efficient for them) strategy. So it could be that the set of the four strategies compared in the paper is perfectly adequate and sufficient to explain the impact of the word blindness suggestion, but that you would need to pick the right one for each participant--something Highs might be able to do spontaneously under hypnosis. This would not be totally surprising, considering how hypnosis is used in clinical settings. Note that it would not be hard to test that with the present data set: select the best strategy for each participant (the one that reduces the Stroop effect the most and speeds up incongruent trials), and redo the analysis on this idiosyncratic strategy. It would also be interesting to check whether the strategy thus selected is the one that is subjectively easier to apply. (Indeed, one may proceed based on this subjective judgment: select the idiosyncratic strategy based on the subjective judgment at the end of each block).

Reply #40

The Reviewer is correct that we did not include an inferential test investigating this issue, we merely speculated about the topic. We have now made up for this by running the second statistical analysis recommended by the Reviewer (selection of the “best” strategy based on subjective judgement of strategy usage). Note that the first proposed analysis would be cherry picking and so cannot provide a strict test of the idea that people differ in the efficiency of their strategy usage. Picking the “best” strategy based on RTs and then analysing the efficiency of the strategies by RTs introduces the problem of selection bias.

The results are interesting. We found that no one preferred the single-letter focus strategy, and it seems that the combination of the looking-away and blurring strategies is not sufficient to account for the word blindness suggestion as they cannot reduce incongruent RTs (data of Experiment 2). The combination of the goal-maintenance, looking-away and blurring strategies (data of Experiment 1) can decrease Stroop interference, but the evidence regarding the reduction of incongruent RTs is insensitive, and so the issue hangs in the balance. We report these results in the Supplementary Materials and we discuss the implications in the General Discussion.

Comment #41

At times, it seems that the authors tend to treat the cold-control theory of hypnosis as a fact. For instance, p. 30: "Nonetheless, none of these strategies should be considered as likely candidates for being the underlying mechanism of the word blindness suggestion, as they did not meet other criteria, such as reducing the RT of incongruent trials." This conclusion holds under the cold control theory, but not necessarily otherwise: suppose for instance that highs after hypnotic induction do modify their first order processes, and not simply their metacognitive access to them (contra Cold-Control). Then, one could imagine that in this state they discover how to blur their vision in a much more efficient way than without hypnosis (in addition to doing that involuntarily). We would then have the same strategy (blurring), but performed at various levels of efficiencies.

Reply #41

See our Reply to Comment #22

Comment #42

p. 35 l. 39 I don't see the motivation for the reservation here expressed: "it still needs to be explored whether those from the full spectrum of hypnotisability can use this strategy to alleviate the interference, and whether the influence of imagination may generalise to other cognitive tasks, such as the flanker (Eriksen & Eriksen, 1974) or Simon task (Simon & Wolf, 1963)". The first part might be needed _under the assumption of the cold control theory_, but again, if one relaxes the constrain of cold control, it might simply be the case that Highs are able to achieve this feat of imagination, while Lows and Mediums would not. As for the last part, it seems not quite relevant to the precise discussion about the Stroop case.

Reply #42

We agree with the reviewer regarding the second point, we removed the references to the Simon and Flanker tasks.

The exploration whether lows and mediums can reduce Stroop interference by voluntarily imagining the words as meaningless is relevant to all theories of hypnosis (to those theories as well that predict a relationship between imagination and hypnotisability). We slightly reworded the sentence to clarify that this exploration is a interesting test of the generalisability of the strategy.

Comment #43

p. 4 l. 18 "the by the"

p. 6 l. 34 McLeod

p. 16 ll 12- 15 trailing zeros could be removed.

p. 26 l. 30 "Table 2" -> 3?

Reply #43

Corrected. We kept the trailing zeros, so all effect sizes are reported to the same decimal point.

Comment #44

p. 27 Figure 3 caption: the "Stroop interference scores" are computed with respect to both incongruent conditions, or only for the traditional semantic + response conflict condition?

Reply #44

They represent the same, traditional Stroop interference (incongruent RT – neutral RT) as we measured in the first experiment and is typically measured in the experiment testing the word blindness effect. We have now made this clear in the figure caption.

Appendix B

Dear Dr Denes Szucs and Essi Viding,

My co-authors and I are happy to submit a revised version of our manuscript "Strategies that reduce Stroop interference" (ID RSOS-202136) for publication in Royal Society Open Science.

We would like to thank you, and the reviewers again for their constructive suggestions for improvement and for the acceptance of the manuscript. Below, in bold, is a point-by-point response to how we have changed the manuscript in order to accommodate the recommendations. We also highlighted all changes in our Manuscript and Supplementary Materials.

Kind regards,
Bence Palfi, on behalf of the authors

Reviewer comments to Author:

Reviewer: 4

Comments to the Author(s)

This aims to uncover the mechanisms of word blindness – a hypnotic phenomenon where high hypnotisable individuals can reduce the Stroop interference effect following a posthypnotic suggestion to experience word stimuli as gibberish. Here, the authors propose to test whether different explicit intentional strategies can replicate the pattern we observe during word blindness. This study present interesting results that accounts for potential explanations of this phenomenon, and thereby test a central assumption of cold control theory – namely that high hypnotisable individuals produce word blindness by the voluntarily implementing one such strategy, while hypnosis yields poor metacognitive appraisal of their intentions to use this strategy.

This manuscript has already been reviewed. Here, I was asked to examine the revisions considering the comments made during the previous round of review. I only highlight the comments and revisions that I find the most relevant.

In my view, the authors addressed all the reviewers' comments to satisfaction.

Aside from the reviewers' comments, my only remark concerns one limitation to the overall rationale of the study with respect to CCT. Although this is a very minor point given the outcome, I nevertheless wonder if the current experiment provides a robust test for CCT. Given that Stroop interference suppression occurs mostly in highs and hardly in lows, if one strategy would have been a suitable candidate to explain word blindness, why would having poor metacognitive representations of intentions, per CCT, explains that highs are more likely adopt this strategy compares to low? Why having good metacognitive representations of intentions would prevent lows from behaving similarly to the highs?

Reply

We thank the reviewer for their work. Importantly, CCT does not claim that highs compared to lows are more likely to adopt a strategy to respond to a suggestion because of the metacognitive differences between highs and lows. As we argue in the last paragraph of the intro, this difference can be easily attributed to variability in motivation (i.e., lows being

unmotivated to create the experience described in the suggestion). The critical prediction of CCT is that when lows and highs adopt the same strategy, they should be able to utilise it to the same extent (i.e., same level of reduction in Stroop interference) even if their metacognitive experiences are different, because, according to CCT, the only difference between them lies in their metacognitive abilities, not in their ability to adopt and utilise strategies. This prediction is tested with Crucial test 3: if highs can use any of the strategies better than lows to reduce Stroop interference, and this difference cannot be attributed to divergence in motivation (or expectations) then the pure metacognitive account of CCT is disconfirmed. To ensure that this argument clearly delivered, we have added a paragraph to the General discussion (p. 37).

Reviewer 1

Comment #1 and #2. Ok

Reply

NA

Comment #3. I agree with the authors that the work from Augustinova and Ferrand (2012) undermines accounts of word blindness arguing for the suppression of semantic processing during the Stroop task. And while evidence tends towards the null hypothesis in both experiments regarding suppression of the semantic Stroop, none offer strong evidence, per the usual $<.33$ threshold adopted by most researchers in the field. Since they went to the trouble of estimating it, I recommend for authors to add the Bayes factors to the manuscript to provide all the information available to the reader.

Reply to Comment #3

We have now included the result of our Bayesian analyses in a footnote (#6). We have also combined the evidence from these two experiments and calculated a meta-analytic Bayes factor, which we report in the same footnote. The meta-analytic Bayes factor is 0.29, indicating substantial evidence for the null.

Comment #4. Again, I concur with the authors, word blindness has already been replicated across several reports and different labs.

Reply

NA

Comment #5. I agree with the authors. The relative differences between conditions we are investigating appears to make this comment somewhat irrelevant, unless there is more to the contingency effect that may interaction with strategies – then it is not all else being equal.

Reply

NA

Reviewer 2

Comment #6. I agree with the authors. This assessment is relevant insofar that variation of

the Stroop interference effects across strategies and hypnotisability has important theoretical implications for CCT.

Reply

NA

Comment #7 to #14. Ok

Reply

NA

Comment #15. Again, I would recommend for this analysis to be included in the manuscript or supplementary material.

Reply to Comment #15

We have now included these analyses in the supplementary materials as part of the carry over effect analyses.

Comment # 16 to # 19. Ok

Reply

NA

Comment #20. The decision to present the reduction of Stroop interference from baseline across strategies addressed the reviewer's about within-subject assessment.

Reply

NA

Comment # 21. Ok

Reply

NA

Reviewer #3

Comment #22. While the authors have made the underlying assumptions of CCT more explicit, the reviewer's comment brings to light a limitation to the rationale of the study. If one these strategies were to replicate the pattern in word blindness, the authors never address how CCT would account for the fact that low hypnotisable individuals fail to intentionally engage in the same strategies as high hypnotisable individuals? I fail to grasp how having accurate second-order representations of intention for lows (as opposed to inaccurate highs) account for the fact that word blindness only occurs for highs? This point comes to be irrelevant considering the results, yet remains pertinent for the rationale of the study.

Reply to Comment #22

See our reply to the comment of Reviewer 4 in which we argue that difference between lows and highs in the extent to which they reduce Stroop interference in a hypnotic context may be due to differences in motivation to engage in strategies in a hypnotic context. We now describe this idea in the general Discussion (p. 37.).

Comment #23. Ok

Reply

NA

Comment #24. Instead of adding the point about counterfactual in a footnote, I would include it in the main body of text since it hardly seems trivial and directly pertains to the main question of this research.

Reply to Comment #24

We have now moved the footnote to the main body of the text.

Comment #25 and #26. Ok

Reply

NA

Comment #27. Again, this footnote might be easily missed.

Reply to Comment #27

In the previous revision, we added this piece of information to the main body of the text in the stimuli subsection, but we have forgotten to correct our reply. Thank you again, for this comment.

Comment #28. I agree with the authors. The statistical tests already directly address the hypothesis. Additional tests about overall slowing down or speeding up are secondary with respect to the main hypothesis. Lastly, the omnibus test would seem redundant and uninformative at this point.

Reply

NA

Comment #29 and #30. Ok

Reply

NA

Comment #31. If they have not done so already, the authors should add this information to the methodology to help with future replications.

Reply to Comment #31

We have now added this to the procedure subsection.

Comment #32 to #37. Ok

Reply

NA

Comment #38. Similar comment as before: Adding this information to supplementary material may provide helpful information to researchers interested in using this method.

Reply to Comment #38

We agree with the reviewer that reporting these results is important. In fact, the corresponding analyses were reported in the main manuscript as part of an outcome neutral test (#2). However, we see now that we haven't made this clear in our response to the reviewer's comment. We have also added the idea that not using equally saturated and luminous colours in the two sets may create a larger interference effect in one of them (p. 26).

Comment #39. While this point is well taken with respect to CCT, knowing whether high hypnotisable individuals are better at implementing certain strategies could have been informative if these strategies offered a potential explanation to word blindness. The idea that at least some highs possess attentional abilities that could ease the implementation for some of these strategies speaks to a whole body of literature. Still, I largely agree with the authors that testing the relationship between the SWASH scores and self-reported ease of strategy implementation is secondary with respect to the main goal of the study, and hardly seem relevant given that none of these strategies represent valid candidate for explaining word blindness and the absence of correlation between SWASH scores and reduction of interference effects across strategies.

Reply

NA

Comment #40 to #44. Ok

Reply

NA